# WAPITI: A Watermark for Finetuned Open-Source LLMs

## ABSTRACT

Watermarking of large language models (LLMs) generation embeds an imperceptible statistical pattern within texts, making it algorithmically detectable. Watermarking is a promising method for addressing potential harm and biases from LLMs, as it enables traceability, accountability, and detection of manipulated content, helping to mitigate unintended consequences. However, for open-source models, watermarking faces two major challenges: (i) incompatibility with fine-tuned models (ii) vulnerability to fine-tuning attacks. In this work, we propose **WAPITI**, a new method that transfers watermarking from base models to fine-tuned models through parameter integration. To the best of our knowledge, we propose the first watermark for fine-tuned open-source LLMs that preserves their fine-tuned capabilities. Furthermore, our approach offers an effective defense against fine-tuning attacks. We test our method on various model architectures and watermarking strategies. Results demonstrate that our method can successfully inject watermarks and is highly compatible with fine-tuned models. Additionally, we offer an in-depth analysis of how parameter editing influences the watermark strength and overall capabilities of the resulting models. [1]

## 1 INTRODUCTION

As large language models (LLMs; Touvron et al., 2023; OpenAI et al., 2024) have been integrated into numerous workflows and play an increasingly significant role in everyday life, controlling these LLMs to prevent potential harm has become even more urgent. Watermarking offers a viable solution by embedding traceable information in model outputs. It enables the identification of LLM-generated content and can be used to trace back to the source model, serving as a methodological foundation for regulatory oversight of language models.

The vast majority of the prior work on watermarks has focused on closed-source models (Kirchenbauer et al., 2024a; Aaronson, 2023; Kuditipudi et al., 2024), which are black boxes for users. However, with the growing capabilities of open-source models (Touvron et al., 2023; Biderman et al., 2023), the need for oversight of open-source models has become equally important. In other words, effective watermarking regulation must take both closed-source and open-source models into account to ensure comprehensive oversight and accountability.

Open-source models release their full parameters to users, and users can fully customize the generation process. Therefore, users can simply choose an unwatermarked decoding algorithm to evade watermarking, thereby invalidating existing decoding-based watermarking methods. Gu et al. (2024) proposed a *parameter-based* method that distills the model using watermarked generations. This process, referred to as *watermark distillation*, ensures that the watermarks are retained within the model parameters, preventing users from easily removing them.

However, we observe that this method (Gu et al., 2024) would impair the fine-tuned capabilities of models, revealing it is not compatible with fine-tuned models. Additionally, watermark distillation incurs significantly higher computational costs compared to typical fine-tuning. Furthermore, a severe weakness of parameter-based watermarks is their vulnerability to *fine-tuning attacks*, where malicious users fine-tune the watermarked models with unwatermarked datasets to eliminate their

---

[1]The model and corresponding code will be released upon publication.

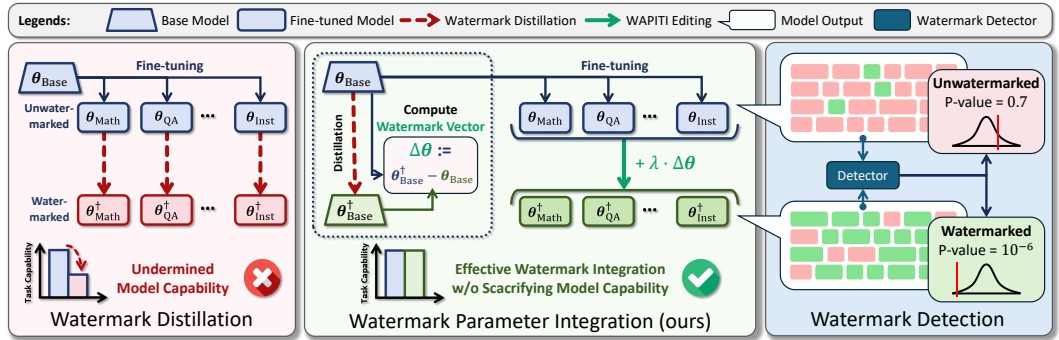

Figure 1: Previous parameter-based watermarking (left) uses distillation which would impair models' fine-tuned capabilities. WAPITI (middle) uses watermark-related parameters to transfer watermarking from the base model to fine-tuned models. This method can preserve fine-tuned model capabilities and meanwhile enables them to generate watermarked texts where the green tokens indicate the watermarked tokens (right).

watermarking. As noted by Gu et al. (2024), as few as 500 steps of fine-tuning attack can remove the watermark from models. (See Table 1 for overall comparison).

To address these limitations, we propose a new train-free[2] watermarking strategy that transfers watermarks from base models to fine-tuned models (**WAPITI**, **WA**termark **P**arameter **I**n**T**egrat**I**on) as shown in Figure 1. We discover that watermarking bears a similar effect on the output distribution of both base models and fine-tuned models. The core of our method involves embedding watermarks into models through direct parameter editing, ensuring compatibility with fine-tuned models. Most importantly, **WAPITI** effectively defends against fine-tuning attacks by binding watermarking with the fine-tuning capabilities of the model.

Our main contributions are as follows:

- **Problem.** We identify the incompatibility between current parameter-based watermarking methods and fine-tuned models. Distillation leads to a rapid degradation of fine-tuning capabilities and fails to effectively apply watermarking to models.
- **Method.** To the best of our knowledge, we propose the first watermarking for fine-tuned models (**WAPITI**) based on the fact that watermarking causes aligned distribution shift in both base models and fine-tuned models.
- **Analysis.** We analyze the relationship between watermarking parameters and model performance, revealing how parameter-editing strength affects final outcomes. Furthermore, we establish the relationship between watermarks and the utility of WAPITI from a learnability perspective.
- **Evaluation.** WAPITI achieves high detectability with an AUROC of 0.92 while maintaining near-identical performance on fine-tuning benchmarks for both the Llama-2-7B and Pythia-1.4B families, demonstrating its strong effectiveness and generality.

## 2 PRELIMINARY

### 2.1 DECODING-BASED WATERMARKING

Large Language Models are generally neural networks based on the transformer architecture, denoted as $f_{\boldsymbol{\theta}} : \mathcal{V}^* \to \Delta(\mathcal{V})$, which maps a given prefix string $\boldsymbol{x} \in \mathcal{V}^*$ to a probability distribution over the vocabulary $\Delta(\mathcal{V})$ for predicting the next token, denoted as $f_{\boldsymbol{\theta}}(\cdot \mid \boldsymbol{x})$. The generation process involves two main steps: *logit generation* followed by *token sampling* (Vaswani et al., 2023).

Decoding-based watermarks are embedded in either stages of generation with the aim of guiding the output distribution toward a targeted direction, incorporating traceable information for detection.

---

[2]"Train-free" means that applying WAPITI to fine-tuned models does not involve any additional model training.

| Method | Closed-source | Open-sourced | | Open-sourced Application | |
|--------|---------------|--------------|--------|--------------------------|--------------|
| | LLMs | Base LLMs | Fine-tuned LLMs | Efficiency | Vulnerability |
| Decoding-based | ✓ | ✗ | ✗ | $\mathcal{C}_{FT}$ | Fine-tuning Attack |
| Distillation-based | N/A | ✓ | ✗ | $\mathcal{C}_{FT}/N$ | Robust to Fine-tuning |
| **WAPITI** | N/A | N/A | ✓ | N/A | N/A |

Table 1: A taxonomy of LLM watermarking. "N/A" indicates that the method is not designed for the corresponding setting. And $\mathcal{C}_{FT}$ indicates the computation cost of watermark distillation. $N$ indicates the number of model of the same type in that WAPITI only requires one watermark distillation to watermarking all models of the same type.

For instance, KGW (Kirchenbauer et al., 2024a) increases the frequency of specific tokens during the generation process, and the detector identifies the origin of a text based on the occurrence rate of these tokens. More specifically, a watermarking algorithm $\mathcal{W}$ employs a watermark key $\phi$ to modify the original next-token distribution $f_{\boldsymbol{\theta}}(\,\cdot\mid\boldsymbol{x})$ into a watermarked version. The watermark detector $\mathcal{D}$, using the same watermark key $\phi$, can then retrieve the embedded watermark information. In general, given a text $x$ and a watermark key $\phi$, the detector $\mathcal{D}$ calculates a p-value for the null hypothesis that the text $x$ is unrelated to $\mathcal{W}$ and $\phi$. A text is classified as model-generated if its p-value falls below a predefined threshold.

The key evaluation metrics of watermarking are: (i) **Detectability:** The watermark must ensure that all content generated by the model can be reliably detected by the detector. (ii) **Utility:** The integration of the watermark should not significantly interfere with the original capabilities of the model. (iii) **Security:** The watermark should ensure that its hidden pattern within the text is difficult to remove unless a substantial portion of the model output is significantly altered. And for open-source models, the watermark cannot be removed without impairing their capabilities.

**Logit-based: KGW** is a watermarking strategy applied directly to output logits of the model (Algorithm 2 in Kirchenbauer et al. (2024a)). During the next token generation, the vocabulary is pseudorandomly split into green and red lists based on the previous $k$ tokens. When $k = 0$ (Zhao et al., 2023), the green and red lists are fixed, and when $k \geq 1$, the lists are determined by the previous context. The green list contains $\gamma \in (0, 1)$ proportion of the entire vocabulary, and an additional watermark shift $\delta$ is added to the logits of the tokens in the green list. This increases the probability of the green tokens being selected in the final generation. During detection, the p-value is calculated by checking whether the proportion of green list tokens exceeds the predefined $\gamma$.

**Sampling-based: AAR** is the Gumbel softmax scheme from Aaronson (2023), which is a special sampling strategy. When generating $x_i$, it hashes the previous $k$ tokens using the key $\phi$ to generate a pseudorandom score sequence $\boldsymbol{r}_i$ for the entire vocabulary $\mathcal{V}$ where $\boldsymbol{r}_i \in \mathbb{R}^{|\mathcal{V}|}$ whose entries are uniformly distributed in $[0, 1]$. Given the probability distribution $\boldsymbol{p}_i \in \Delta(\mathcal{V})$ of the next token $x_i$, AAR uses Gumbel-Max sampling strategy: $x_i = \arg\max_{j \in |\mathcal{V}|}(\log p_{i,j} - \log(-\log r_{i,j}))$ (Cane & Luce, 1960), which introduces some randomness into the sampling stage by adding Gumbel noise $\boldsymbol{r}_i$. This sampling strategy would result in watermarked texts having comparative higher score sums. During detection, a larger score sum corresponds to a lower p-value against the null hypothesis.

## 2.2 WEIGHT-BASED WATERMARKING

Since the weights of open-source models are fully released, users can modify the decoding method or apply any post-processing to the logits, making decoding-based watermarks easy to remove. The most feasible approach[3] for watermarking is to embed the watermark into the model parameters, enabling LLMs to generate watermarked text under natural sampling distribution. Current research (Gu et al., 2024) has shown that LLMs can learn watermarks via distillation and generate detectable watermarked texts. By using decoding-based watermark strategies to generate watermarked texts as distillation data, Gu et al. (2024) has verified the learnability of multiple watermarks on Llama-2-7B and Pythia-1.4B models. However, we found that this parameter-based method is specifically de-

---

[3]To the best of our knowledge, this is the only approach for watermarking open-source LLMs that cannot be easily removed by users.

signed for base LLMs. In the fine-tuning setting, it significantly impairs the fine-tuned capabilities, as we will demonstrate in § 3.1.

# 3 METHOD

## 3.1 MOTIVATING STUDY

**Limitation of current weight-based watermarking.** The current weight-based method enables the base model to generate watermarked texts via distillation. In this paper, we explore whether the distillation-based approach is compatible with fine-tuned models. Specifically, we ask: can watermark distillation retain the fine-tuned capabilities of the model while embedding the watermark into the fine-tuned model? To address this question, we conduct a preliminary experiment.

To obtain a watermarked fine-tuned model using watermark distillation, there are three possible approaches: (i) Distilling a fine-tuned model with watermarked content, (ii) Fine-tuning a distilled model that already contains a watermark, or (iii) Fine-tuning a base model using a watermarked fine-tuning dataset. We use math-fine-tuned Llama-2-7B and decoding-based watermarking strategies to obtain a watermarked math model. Detailed experimental setups can be found in Appendix A.

Figure 2 compares the watermark detectability (measured by p-value) and fine-tuning utility of the resulting model from all three different approaches. The utility of the models on GSM8K drops sharply to nearly zero, and the output text shows poor detectability, with a p-value close to the baseline of 0.5.

To better understand this phenomenon, we further analyze the three approaches. The first two methods both involve two-phase fine-tuning, which, as studied in previous research, can lead to capability degradation or catastrophic forgetting (Wang et al., 2023).

For the third method, we believe it holds the most potential to enable the base model to learn mathematical capabilities while embedding the watermark content. Therefore, we focus on analyzing the distillation data generated by the math-fine-tuned model and identify two main reasons: (i) The quality of the watermarked math data is inferior to that of the original fine-tuning dataset. Table 4 presents several samples from the original benchmark dataset alongside the answers generated by the math model. Although the generated answers might still be correct, they often contain flawed procedures or random repetitive sequences. Such data can confuse the model and result in a performance decline. (ii) The quantity of watermarked math data is insufficient for the student model to learn the watermark effectively. As noted in Gu et al. (2024), approximately 1.3 million samples are required for a distilled model to internalize the watermark. With only 7.3k samples in the GSM8K training split and further filtering due to the 40% accuracy of model, our final dataset was just 0.6% of the required size. The insufficient distillation data makes the generated outputs lack detectability.

In a nutshell, our experiments demonstrate that current distillation-based watermarking is incompatible with fine-tuned models. This is primarily due to the small size of most fine-tuning datasets, which are insufficient for distillation. Additionally, the quality of watermarked samples deteriorates compared to the original ones, leading to a decline in the fine-tuning capabilities of model.

**Universal distribution shift from watermarking.** The primary issue with the current weight-based method is the distillation phase, which underscores the need for a train-free approach to watermark fine-tuned models. To this end, we aim to investigate whether there are any similarities between the base models and fine-tuned models when watermarked.

To be specific, we analyze the $n$-gram distribution in the watermarked outputs of both the base and fine-tuned models. According to watermarking schemes, $n$-gram could be the smallest meaningful unit, making them a natural starting point. Check Appendix C for detailed justification.

Our experiment compares the $n$-gram distribution similarities between unwatermarked and watermarked texts generated by the base model and fine-tuned model, respectively. We used Llama-2-7B and the math fine-tuned Llama-2-7B (Agarwalla et al., 2024) to generate 640k samples with and without watermarking. The watermark used were kgw-k1-gamma0.25-delta2 (Kirchenbauer et al., 2024a) and aar-k2 (Aaronson, 2023).

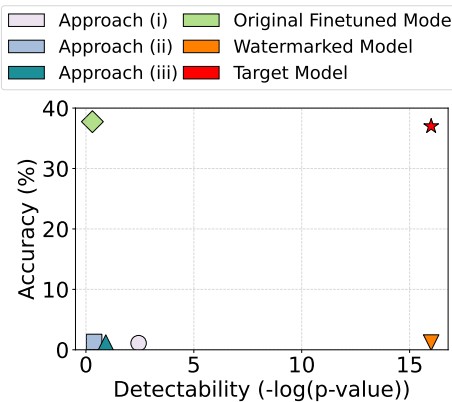

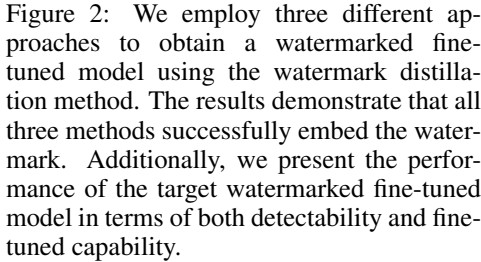

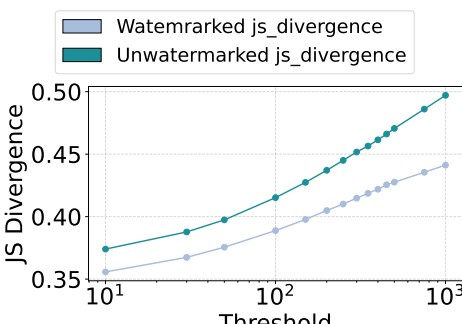

Figure 2: We employ three different approaches to obtain a watermarked fine-tuned model using the watermark distillation method. The results demonstrate that all three methods successfully embed the watermark. Additionally, we present the performance of the target watermarked fine-tuned model in terms of both detectability and fine-tuned capability.

Figure 3: JS divergence between watermarked and unwatermarked $n$-grams across different sifting thresholds. Results show that watermarking has partially aligned the output distributions of the base model and fine-tuned model.

We tokenize all generated text into $n$-grams, where $n$ is determined by the number of tokens used to compute the watermarked next-token probability, as mentioned in § 2.1. We then calculate the Jensen–Shannon (JS) divergence (Lin, 1991) between the watermarked and unwatermarked $n$-grams. To reduce noise, we filter out $n$-grams whose frequencies are below a threshold.

The results, shown in Figure 3, indicate that the JS divergence is consistently smaller for watermarked $n$-gram compared to unwatermarked $n$-gram, which suggests that the distribution of watermarked $n$-gram is more similar between base models and fine-tuned models. This indicates that watermarking distorts the output of both the base and fine-tuned models in similar ways by increasing the frequency of watermarked $n$-grams in the final generation.

## 3.2 WATERMARK PARAMETER

In this section, we focus on deriving the watermarked parameters of fine-tuned models. As mentioned in §2.1, watermarks only perturb the next-token generation $x_t$ according to previous $k$ tokens $x_{t-k}, \cdots, x_{t-1}$ and watermark key $\phi$, so that watermark perturbation in next-token probability $f_{\boldsymbol{\theta}}(\boldsymbol{x})$[4] remains the same across different models, where $\boldsymbol{x}$ is the input prompt. We denote the watermark perturbation as $\delta \cdot g(\boldsymbol{x})$, where $\delta$ represents the intensity of the shift, analogous to the watermark shift $\delta$ in KGW and $g(\boldsymbol{x})$ is analogous to the mask of green list in KGW watermarking that indicates which part of vocabulary will be applied watermark shift. According to experiments in C, we observe that model parameters can learn watermarking. Let $\boldsymbol{\theta}_{\text{Base}}, \boldsymbol{\theta}_{\text{Base}}^{\dagger}$ represent parameters of the base model and the watermark-distilled base model respectively. So we have:

$$f_{\boldsymbol{\theta}_{\text{Base}}^{\dagger}}(\boldsymbol{x}) = f_{\boldsymbol{\theta}_{\text{Base}}}(\boldsymbol{x}) + \delta_{\text{Base}} \cdot g(\boldsymbol{x}). \tag{1}$$

Similarly, we use $\boldsymbol{\theta}_{\text{FT}}$ and $\boldsymbol{\theta}_{\text{FT}}^{\dagger}$ to represent the parameters of the fine-tuned (FT) models, as well as its watermark-distilled counterpart respectively. Our ultimate goal is, given an unwatermarked $\boldsymbol{\theta}$, to find the parameter $\boldsymbol{\theta}_{\text{FT}}^{\dagger}$ such that:

$$f_{\boldsymbol{\theta}_{\text{FT}}^{\dagger}}(\boldsymbol{x}) = f_{\boldsymbol{\theta}_{\text{FT}}}(\boldsymbol{x}) + \delta_{\text{FT}} \cdot g(\boldsymbol{x}), \tag{2}$$

where $\delta_{\text{FT}}$ is a hyperparameter that controls the watermark detectability.

---

[4]For brevity, we identify the next-token probability predictor $f_{\boldsymbol{\theta}}(\cdot \mid \boldsymbol{x}) : \mathcal{V} \to \mathbb{R}$ as a vector $f_{\boldsymbol{\theta}}(\boldsymbol{x}) \in \Delta(\mathcal{V})$.

Let $\Delta\boldsymbol{\theta}_{\text{Base}} := \boldsymbol{\theta}^{\dagger}_{\text{Base}} - \boldsymbol{\theta}_{\text{Base}}$ and $\Delta\boldsymbol{\theta}_{\text{FT}} := \boldsymbol{\theta}^{\dagger}_{\text{FT}} - \boldsymbol{\theta}_{\text{FT}}$ denote the parameter differences introduced by watermark distillation for the base and fine-tuned models, respectively. We can eliminate $g(\boldsymbol{x})$ by substituting $\boldsymbol{\theta}_{\text{Base}}, \boldsymbol{\theta}^{\dagger}_{\text{Base}}$ into Eq. (1) and rearranging it as a Taylor expansion. $\delta_{\text{Base}}$ denotes the watermark shift of base model:

$$g(\boldsymbol{x}) = \frac{1}{\delta_{\text{Base}}}\left(f_{\boldsymbol{\theta}^{\dagger}_{\text{Base}}}(\boldsymbol{x}) - f_{\boldsymbol{\theta}_{\text{Base}}}(\boldsymbol{x})\right) = \frac{1}{\delta_{\text{Base}}}\langle\nabla_{\boldsymbol{\theta}}f_{\boldsymbol{\theta}_{\text{Base}}}(\boldsymbol{x}), \Delta\boldsymbol{\theta}_{\text{Base}}\rangle + O(\|\Delta\boldsymbol{\theta}_{\text{Base}}\|^2). \quad (3)$$

Furthermore, we observe in Appendix F.3, that the parameter difference between the fine-tuned model and the base model, $\boldsymbol{\theta}_{\text{FT}} - \boldsymbol{\theta}_{\text{Base}}$, is approximately orthogonal to the parameter difference caused by watermarking, $\boldsymbol{\theta}^{\dagger}_{\text{Base}} - \boldsymbol{\theta}_{\text{Base}}$:

$$\langle\boldsymbol{\theta}_{\text{FT}} - \boldsymbol{\theta}_{\text{Base}}, \boldsymbol{\theta}^{\dagger}_{\text{Base}} - \boldsymbol{\theta}_{\text{Base}}\rangle \approx 0. \quad (4)$$

Let $\otimes$ denote the tensor product between differentiation operators, and let $\times_1, \times_2$ denote the mode-1 and mode-2 tensor–matrix product, respectively. Let $\boldsymbol{H}_{\text{Base}}(\boldsymbol{x}) := \nabla_{\boldsymbol{\theta}}\otimes\nabla_{\boldsymbol{\theta}}f_{\boldsymbol{\theta}_{\text{Base}}}(\boldsymbol{x})$ be the Hessian. As shown in prior studies, every channel of $\boldsymbol{H}_{\text{Base}}(\boldsymbol{x})$ is approximately the identity matrix $\boldsymbol{I}$ (Jiao et al., 2024; Yang et al., 2024). Combining it with our observation in Eq. (4), we hypothesize that:

$$\boldsymbol{H}_{\text{Base}}(\boldsymbol{x}) \times_1 (\boldsymbol{\theta}_{\text{FT}} - \boldsymbol{\theta}_{\text{Base}}) \times_2 (\boldsymbol{\theta}^{\dagger}_{\text{Base}} - \boldsymbol{\theta}_{\text{Base}}) \approx 0. \quad (5)$$

The first-order Taylor expansion of $\nabla_{\boldsymbol{\theta}}f_{\boldsymbol{\theta}_{\text{FT}}}(\boldsymbol{x})$ around $\boldsymbol{\theta} = \boldsymbol{\theta}_{\text{Base}}$ is:

$$\nabla_{\boldsymbol{\theta}}f_{\boldsymbol{\theta}_{\text{FT}}}(\boldsymbol{x}) = \nabla_{\boldsymbol{\theta}}f_{\boldsymbol{\theta}_{\text{Base}}}(\boldsymbol{x}) + \boldsymbol{H}_{\text{Base}}(\boldsymbol{x}) \times_1 (\boldsymbol{\theta}_{\text{FT}} - \boldsymbol{\theta}_{\text{Base}}) + O(\|\boldsymbol{\theta}_{\text{FT}} - \boldsymbol{\theta}_{\text{Base}}\|^2), \quad (6)$$

$$\boldsymbol{H}_{\text{Base}}(\boldsymbol{x}) \times_1 (\boldsymbol{\theta}_{\text{FT}} - \boldsymbol{\theta}_{\text{Base}}) \approx \nabla_{\boldsymbol{\theta}}f_{\boldsymbol{\theta}_{\text{FT}}}(\boldsymbol{x}) - \nabla_{\boldsymbol{\theta}}f_{\boldsymbol{\theta}_{\text{Base}}}(\boldsymbol{x}). \quad (7)$$

Next, substituting Eq. (7) into Eq. (5), we find that the gradient difference between the fine-tuned and base models, when multiplied by the watermarked parameter difference of base model, is approximately zero:

$$(\nabla_{\boldsymbol{\theta}}f_{\boldsymbol{\theta}_{\text{FT}}}(\boldsymbol{x}) - \nabla_{\boldsymbol{\theta}}f_{\boldsymbol{\theta}_{\text{Base}}}(\boldsymbol{x}))\Delta\boldsymbol{\theta}_{\text{Base}} \approx 0. \quad (8)$$

By rearranging Eq. (8), we conclude that the gradients of the fine-tuned and base models are approximately equal when applied to the watermarked parameter difference:

$$\nabla_{\boldsymbol{\theta}}f_{\boldsymbol{\theta}_{\text{FT}}}(\boldsymbol{x})\Delta\boldsymbol{\theta}_{\text{Base}} \approx \nabla_{\boldsymbol{\theta}}f_{\boldsymbol{\theta}_{\text{Base}}}(\boldsymbol{x})\Delta\boldsymbol{\theta}_{\text{Base}}. \quad (9)$$

In this way, we obtain the relationship between the gradient of the fine-tuned model and base models. And we now proceed to derive our target $f_{\boldsymbol{\theta}_{\text{FT}}}(\boldsymbol{x})$. First, by substituting $g(\boldsymbol{x})$ from Eq. (3) into Eq. (2):

$$f_{\boldsymbol{\theta}^{\dagger}_{\text{FT}}}(\boldsymbol{x}) = f_{\boldsymbol{\theta}_{\text{FT}}}(\boldsymbol{x}) + \left(\frac{\delta_{\text{FT}}}{\delta_{\text{Base}}}\langle\nabla_{\boldsymbol{\theta}}f_{\boldsymbol{\theta}_{\text{Base}}}(\boldsymbol{x}), \Delta\boldsymbol{\theta}_{\text{Base}}\rangle + O(\|\Delta\boldsymbol{\theta}_{\text{Base}}\|^2)\right). \quad (10)$$

We define $\lambda_{\text{FT}} = \frac{\delta_{\text{FT}}}{\delta_{\text{Base}}}$, where $\delta_{\text{FT}}$ is a hyperparameter, making $\lambda_{\text{FT}}$ a tunable factor. Next, we substitute the gradient of base model in Eq. (10) with the gradient of fine-tuned model using Eq. (9):

$$f_{\boldsymbol{\theta}^{\dagger}_{\text{FT}}}(\boldsymbol{x}) \approx f_{\boldsymbol{\theta}_{\text{FT}}}(\boldsymbol{x}) + \langle\nabla_{\boldsymbol{\theta}}f_{\boldsymbol{\theta}_{\text{FT}}}(\boldsymbol{x}), \lambda_{\text{FT}} \cdot \Delta\boldsymbol{\theta}_{\text{Base}}\rangle + O\left(\|\Delta\boldsymbol{\theta}_{\text{Base}}\|^2\right), \quad (11)$$

$$\approx f_{\boldsymbol{\theta}_{\text{FT}} + \lambda_{\text{FT}} \cdot \Delta\boldsymbol{\theta}_{\text{Base}}}(\boldsymbol{x}). \quad (12)$$

We treat Eq. (11) as a Taylor expansion of the next-token probability of the model with respect to its parameters. Based on Eq. (12), we can select:

$$\boldsymbol{\theta}^{\dagger}_{\text{FT}} := \boldsymbol{\theta}_{\text{FT}} + \lambda_{\text{FT}} \cdot \Delta\boldsymbol{\theta}_{\text{Base}}. \quad (13)$$

According to derivation, we propose *WAtermark Parameter InTegratIon* (WAPITI), which integrates watermark-related parameters of base model to fine-tuned models. The algorithm is shown in Alg. 1. WAPITI is compatible with various watermarking strategies: after distilling a base model with the desired watermark (Step 1), the watermark can be seamlessly transferred to fine-tuned models without additional costs (Step 3). This approach provides an efficient and effective solution for regulating open-source models.

---

**Algorithm 1** WAPITI

**Input:** base model parameter $\boldsymbol{\theta}_{\text{Base}}$, fine-tuned model parameter $\boldsymbol{\theta}_{\text{FT}}$, watermark intensity factor $\lambda_{\text{FT}}$

**Output:** watermarked fine-tuned model parameter $\boldsymbol{\theta}^{\dagger}_{\text{FT}}$

1: $\boldsymbol{\theta}^{\dagger}_{\text{Base}} \leftarrow \text{WatermarkDistillation}(\boldsymbol{\theta}_{\text{Base}})$
2: $\Delta\boldsymbol{\theta}_{\text{Base}} \leftarrow \boldsymbol{\theta}^{\dagger}_{\text{Base}} - \boldsymbol{\theta}_{\text{Base}}$
3: $\boldsymbol{\theta}^{\dagger}_{\text{FT}} \leftarrow \boldsymbol{\theta}_{\text{FT}} + \lambda_{\text{FT}} \cdot \Delta\boldsymbol{\theta}_{\text{Base}}$

---

## 4 EXPERIMENT

### 4.1 EXPERIMENTAL SETUP

In this section, we design experiments to evaluate the utility of WAPITI in two key aspects: *watermark strength* and *fine-tuning ability*, tested across various models and watermarking strategies.

**Watermark and hyperparameters.** We experiment with two representative decoding-based watermarks, KGW and AAR, with different hyperparameters. To ensure a fair and consistent comparison, we adopt the same watermarking hyperparameters as used by Gu et al. (2024). Specifically, for KGW, we set $k = \{0, 1, 2\}$, $\gamma = 0.25$, and $\delta = \{1, 2\}$; and for AAR, we use $k = \{2, 3, 4\}$. The coefficient $\lambda_{FT}$ for watermark parameter integration ranges from $[0, 4]$.

**Dataset and model choices.** To ensure the generalizability of WAPITI, we conduct experiments on two widely used LLM families: Llama-2-7B and Pythia-1.4B, which differ in both architecture and parameter Their popularity in the community further ensures that our experiments reflect real-world utility. We utilize the watermark-distilled base models from Gu et al. (2024).

**Evaluation Procedure** We evaluate model generation using samples from the RealNewsLike subset of the C4 dataset (Raffel et al., 2023). Specifically, the evaluation sample size is 5,000, with a 50-token prompt and a sequence length of 200. We use temperature sampling with $t = 1$.

Building on the approach of Gu et al. (2024), we apply deduplication during post-processing to remove repetitive generations, ensuring the validity of the final detectability results.

**Evaluation Metrics** To test the compatibility of WAPITI with fine-tuned models, we focus on three key fine-tuning capabilities: *instruction-following*, *question answering*, and *math*. We will refer to corresponding fine-tuned models as Llama-chat, Llama-QA, Llama-gsm8k, and Pythia-chat in the experiment results. Detailed information on the fine-tuned model selection can be found in Appendix D. The benchmark datasets used are OpenWebText (Gokaslan & Cohen, 2019), MMLU (Hendrycks et al., 2021), and GSM8K (Cobbe et al., 2021), respectively.

### 4.2 EVALUATION METRICS

Following the evaluation methods used in Kirchenbauer et al. (2024a), Kuditipudi et al. (2024), and Gu et al. (2024), we evaluate the models on 5,000 samples drawn from the RealNewsLike subset of the C4 dataset (Raffel et al., 2023). The evaluation includes the following metrics:

**Watermark detectability.** To assess watermark detectability, we compute the median p-value and AUROC (Area Under the Receiver Operating Characteristic Curve), which evaluates the ability to distinguish between watermarked and unwatermarked content. The p-value is computed using the z-score method. A lower p-value indicates stronger watermark detectability. The AUROC is calculated using an equal number of human-generated texts and model-generated watermarked content, both truncated to the same length for consistency.

**Generation quality.** Generation quality is evaluated using two metrics: perplexity and seq-rep-3 (Sequence Repetition for 3-grams). Perplexity provides an overall assessment of the generated text and is calculated using Llama-2-13B. Seq-rep-3 measures repetition by calculating the proportion of repeated trigrams (Welleck et al., 2019).

**Fine-tuning abilities.** To assess whether WAPITI preserves the fine-tuned capabilities of models, we evaluate the performance of WAPITI fine-tuned models on the following benchmarks: i) Question Answering: We use the full MMLU (Hendrycks et al., 2021) dataset to assess the QA ability of models. This dataset contains approximately 14,000 questions from 57 domains. ii) Math: We evaluate the model on the test split of GSM8K (Cobbe et al., 2021), which consists of 1,319 grade-school math word problems designed to assess multi-step reasoning and arithmetic skills.

### 4.3 RESULTS

**Watermarking results.** Table 2 presents the results of the watermark strength and generation quality of the WAPITI model. Since multiple hyperparameter sets were tested for each watermarking strategy, the result table displays the average across all hyperparameter sets for each watermark, with

| Scheme | Model | Watermark Detectibility | | | | Generation Quality | | | |
| --- | --- | --- | --- | --- | --- | --- | --- | --- | --- |
| | | p-value($\downarrow$) | | AUROC($\uparrow$) | | Perplexity($\downarrow$) | | seq-rep-3($\downarrow$) | |
| | | DECO | WAPITI | DECO | WAPITI | DECO | WAPITI | DECO | WAPITI |
| **KGW** | Llama-distilled | $4.2\cdot10^{-25}$ | $3.5\cdot10^{-15}$ | 0.99 | **0.94** | 5.91 | 5.85 | 0.05 | 0.03 |
| | Llama-gms8k | $5.7\cdot10^{-18}$ | $1.3\cdot10^{-12}$ | 0.96 | 0.92 | 4.03 | 4.15 | 0.19 | 0.12 |
| | Llama-chat | $1.9\cdot10^{-8}$ | $7.9\cdot10^{-7}$ | 0.92 | 0.90 | **3.12** | **3.16** | 0.08 | 0.05 |
| | Llama-QA | $5.1\cdot10^{-13}$ | $8.1\cdot10^{-7}$ | 0.96 | 0.91 | 3.50 | 3.44 | 0.08 | 0.04 |
| | Pythia-distilled | $2.6\cdot10^{-12}$ | $6.9\cdot10^{-4}$ | 0.98 | 0.78 | 12.4 | 20.0 | 0.04 | **0.02** |
| | Pythia-chat | $5.3\cdot10^{-11}$ | $1.48\cdot10^{-1}$ | 0.90 | 0.61 | 7.23 | 6.86 | 0.06 | 0.07 |
| **AAR** | Llama-distilled | $4.2\cdot10^{-88}$ | $3.6\cdot10^{-12}$ | **1.00** | 0.80 | 27.1 | 5.18 | 0.05 | 0.06 |
| | Llama-gms8k | $\mathbf{6.3\cdot10^{-92}}$ | $6.2\cdot10^{-8}$ | **1.00** | 0.77 | 9.13 | 3.73 | 0.15 | 0.14 |
| | Llama-chat | $1.6\cdot10^{-57}$ | $7.4\cdot10^{-7}$ | **1.00** | 0.78 | 20.2 | 3.18 | 0.06 | 0.07 |
| | Llama-QA | $5.3\cdot10^{-64}$ | $4.4\cdot10^{-6}$ | **1.00** | 0.78 | 5.9 | 3.45 | 0.06 | 0.07 |
| | Pythia-distilled | $2.0\cdot10^{-73}$ | $\mathbf{7.3\cdot10^{-18}}$ | **1.00** | 0.85 | 10.5 | 10.8 | **0.03** | 0.21 |
| | Pythia-chat | $3.3\cdot10^{-66}$ | $2.08\cdot10^{-1}$ | **1.00** | 0.61 | 10.1 | 9.41 | **0.03** | 0.07 |
| **None** | Base Llama | $4.5\cdot10^{-1}$ | | 0.48 | | 3.14 | | 0.03 | |
| | Base Pythia | $5.6\cdot10^{-1}$ | | 0.49 | | 10.3 | | 0.04 | |

Table 2: Main results for watermark detectability and generation quality of WAPITI and decoding-based watermarks across different strategies. The displayed results represent the average performance, with an integration coefficient of $\lambda_{\text{FT}} = 1$. *DECO* refers to the original decoding-based watermark used as the baseline.

the embedded watermark parameter integration coefficient $\lambda_{\text{FT}}$ fixed to 1.0. Detailed results for each hyperparameter, as well as the full set of results for different values of $\lambda_{\text{FT}}$, along with corresponding analysis, can be found in Appendix G.

The results show that WAPITI effectively transfers the watermark to other models, achieving low p-values and high AUROC scores, indicating strong detectability. Additionally, the generation quality metrics confirm that WAPITI preserves the models' original capabilities. However, the detectability in WAPITI fine-tuned models is slightly lower compared to the watermark-distilled base models, suggesting that some watermarking information is lost during the transfer process.

Of the two watermarks tested, KGW consistently outperforms AAR in watermark transfer, exhibiting higher AUROC scores. This trend is also observed in the watermark-distilled models from Gu et al. (2024), which we partly attribute to the complexity of the AAR scheme, as it combines logits with pseudorandom scores. A more detailed analysis of this difference is provided in Appendix F.1. Comparing the performance across different models, the watermark detectability in Pythia models is lower than in Llama models. Analyzing the generations of Pythia models suggests that this difference is largely due to the models' inherent capabilities. Nevertheless, the parameter integration yields p-values significantly below the baseline of 0.5, indicating that watermarking-related knowledge is still injected to a certain degree.

**Fine-tuned ability results.** Figure 4 compares the fine-tuning performance of WAPITI models with the base model and original fine-tuned models. For both QA and Math tasks, WAPITI models show performance nearly identical to the original fine-tuned models for both KGW and AAR watermarking, demonstrating that WAPITI effectively preserves the models' original capabilities and is fully compatible with fine-tuned models.

Combined with the results from Table 2, we conclude that WAPITI is an effective and efficient watermarking method for fine-tuned models, allowing

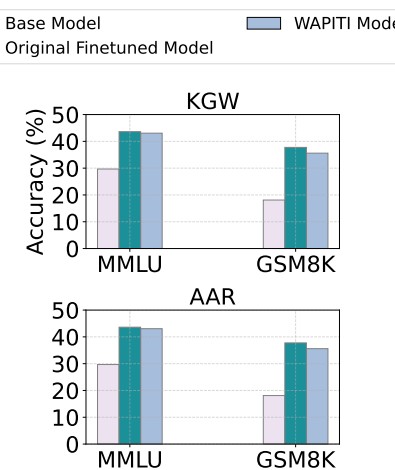

them to retain the watermark while preserving both generation quality and fine-tuned capabilities.

**Robustness to Attacks**

WAPITI embeds watermarks through parameter integration, so the watermark parameters must be kept secret, similar to a secret key. Otherwise, malicious users could directly remove the integrated parameters to invalidate the watermark.

Beyond this direct attack, we evaluate WAPITI's robustness against classical watermark elimination methods, including text edits and changes in decoding parameters. Detailed experimental setups and robustness analyses are provided in Appendix E.

## 4.4 ANALYSIS

In this section, we conduct additional experiments to examine how watermark parameters impact the overall WAPITI models' watermark detectability and capabilities, providing insights into WAPITI for better utilization and future works.

First, we examine how the norm of watermark parameter integration and the hyperparameters of the watermarking schemes impact detectability and generation quality. We vary the coefficient $\lambda_{FT}$ within the range $[0, 4]$ to test WAPITI models' median p-values and perplexity. The results show that $\lambda_{FT}$ regulates the interference between watermarked and model parameters, and detectability strongly correlates with watermark learnability. Full results and analysis are in Appendix F.1.

Second, we evaluate whether WAPITI can defend against fine-tuning attacks by binding fine-tuned capabilities with watermarking. The results show that malicious users would significantly degrade the fine-tuned capabilities of models when attempting to remove the watermark through fine-tuning attacks. Full results can be found in Appendix F.2.

Finally, we analyze the relationship between the fine-tuned and watermarked models at the parameter level using cosine similarity (Ilharco et al., 2023), illustrating how WAPITI remains compatible with fine-tuned models. This analysis also provides strong evidence that watermarked parameters indeed encode knowledge about the watermarking schemes. Full results are detailed in Appendix F.3.

## 5 RELATED WORK

**Text steganography.** Steganography involves embedding information within texts for the purposes of detection or secret communication. Steganography methods can be categorized into *edit-based* and *generative* approaches. Edit-based methods include rule-based transformations (Wilson et al., 2014; Wilson & Ker, 2016), synonym-based substitution (Shirali-Shahreza & Shirali-Shahreza, 2008), and neural network-based transformations (Fang et al., 2017; Abdelnabi & Fritz, 2021; Ueoka et al., 2021). On the other hand, generative methods embed information directly during the text generation process (Ziegler et al., 2019; Dai & Cai, 2019).

**Text watermarking.** Earlier works in text watermarking typically embedded information through post-processing of texts, closely resembling steganography (Venugopal et al., 2011; Yang et al., 2021). More recent studies have shifted towards decoding-based watermarking, hiding information by perturbing the text during the decoding phase (Kirchenbauer et al., 2024b; Aaronson, 2023; Zhu et al., 2024; Krishna et al., 2023; Kuditipudi et al., 2024; Zhao et al., 2023; Christ et al., 2023; Wu et al., 2024; Liu & Bu, 2024; Giboulot & Teddy, 2024; Lu et al., 2024; Ren et al., 2024; Wang et al., 2024). Different watermarking strategies bring various improvements: Takezawa et al. (2023) enhance logit-perturbation, while Hu et al. (2023); Zhao et al. (2024) optimize sampling strategies. Additionally, Lee et al. (2024); Li et al. (2023); Yang et al. (2021) explore code watermarking.

Recent advancements have introduced parameter-based watermarking, which embeds watermarks through distillation (Gu et al., 2024). Other studies focus on investigating typical watermarking behaviors (Luo et al., 2024; Singh & Zou, 2023), and some establish robust statistical frameworks for watermarking (Huang et al., 2024; Li et al., 2024). Surveys provide detailed definitions and

classifications of text watermarking techniques (Jawahar et al., 2020; Liu et al., 2024; Cai et al., 2024), while benchmarks offer comprehensive evaluations of watermarks (Tu et al., 2024).

**Model interventions.** Beyond fine-tuning, researchers have explored parameter-level interventions to modify model behaviors. Key approaches include model patching (Goel et al., 2020; Ilharco et al., 2022; Murty et al., 2022; Sung et al., 2021), parameter editing (Mitchell et al., 2022a;b; Santurkar et al., 2021; Ilharco et al., 2023), and model alignment (Askell et al., 2021; Glaese et al., 2022; Kasirzadeh & Gabriel, 2022). Compared to retraining or fine-tuning, model intervention offers a more efficient way to introduce new capabilities into models.

## 6 CONCLUSION

In this paper, we propose WAPITI, a training-free, parameter-based watermarking scheme designed for fine-tuned open-source models. We evaluate its effectiveness on various model architectures and watermarking strategies. Our method resolves the key technical challenges of applying watermarks to fine-tuned models while retaining the fine-tuned model abilities. Furthermore, we analyze the relationship between parameter integration and the model performance, using cosine similarity analysis to demonstrate that the watermarking parameters encode $n$-gram related knowledge.

Future work could further enhance WAPITI by developing watermarking strategies better suited to watermark transfer or optimizing the watermark distillation process to produce better watermark-distilled base models. Additionally, refining the extraction procedure for watermark parameters could improve the efficiency of watermark transfer. This would also minimize interference with other model parameters, helping to preserve the overall model performance.

## LIMITATION

This study introduces WAPITI and provides theoretical support; however, the derivation relies on assumptions based on several experimental results. Our experiments, which include three different fine-tuning models and two model structures of varying parameter sizes, align with our assumptions. Nonetheless, the generality of these assumptions may require deeper analysis of the experimental results and testing on more diverse datasets to ensure their robustness.

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

# Appendix

## Table of Contents

## A   FINE-TUNED MODELS' WATERMARKING DISTILLATION SETUP

We use Neuralmagic Llama-2-7B-gsm8k (Agarwalla et al., 2024) as both the teacher and student models. Mathematics is selected as the fine-tuned capability, with GSM8K (Cobbe et al., 2021) serving as the fine-tuning dataset.

Due to the low-entropy nature of mathematics questions, which can interfere with the watermarking process, we use Chain-of-Thought (CoT) prompting on the fine-tuning samples to expand the entropy space and improve detectability. Examples of (Question, Answer) pairs are provided in Table 4, ensuring that the model demonstrates both fine-tuned capability and watermarking simultaneously.

For watermarking, we select the schemes `kgw-k0-gamma-0.25-delta-2` and `aar-k3`, as they are relatively easier for the model to learn from. We will now introduce the specific setups for the three different approaches described in § 2.1.

**Distilling a fine-tuned model with watermarked content**    We use a math-fine-tuned model as the student model and Llama 2 7B (Touvron et al., 2023) as the teacher model. The distillation process utilizes the OpenWebText dataset (Gokaslan & Cohen, 2019) for 1,000 steps.

The batch size is set to 64, the sequence length to 512, and the maximum learning rate to $1 \times 10^{-5}$, with a cosine learning rate decay and a linear warmup during the first 200 steps. Each training session takes approximately 2 hours on 4 NVIDIA A100 80GB GPUs.

**Fine-tuning a distilled model that already contains a watermark**    We used a watermarked fine-tuned Llama 2 7B model and fine-tuned it further on GSM8K to enhance its math capabilities. The total training consisted of 129 steps with a batch size of 64 sequences and a sequence length of 256 tokens.

The maximum learning rate was set to $1 \times 10^{-5}$, with a cosine learning rate decay and a linear warmup over the first 20 steps. We employed the AdamW optimizer with $(\beta_1, \beta_2) = (0.9, 0.999)$ and no weight decay. Each training run took approximately 50 minutes on 4 NVIDIA A100 80GB GPUs.

**Fine-tuning a base model using a watermarked fine-tuning dataset.**    First, we generated watermarked samples of 256 tokens using a 50-token prefix from GSM8K as the prompt. These watermarked generations were filtered based on the correctness of their final answers, resulting in 2,632 correct samples, which were used as training data for distilling the Llama-2-7B-gsm8k model.

Next, we fine-tuned Llama-2-7B-gsm8k on the watermarked samples for 3 epochs, with 43 steps per epoch, using a batch size of 64 sequences and a sequence length of 256 tokens. The maximum learning rate was set to $1 \times 10^{-5}$, with a cosine learning rate decay and a linear warmup over the first 20 steps. We used the AdamW optimizer with $(\beta_1, \beta_2) = (0.9, 0.999)$ and no weight decay. Each training run took approximately 60 minutes on 4 NVIDIA A100 80GB GPUs.

In our experiments, we use Math, QA, and instruction-tuning models based on Llama-2-7B. Since Gu et al. (2024) provides pre-watermarked distilled models for Llama-2-7B, the watermarking process for these models incurs no additional training cost. Even if pre-watermarked models are unavailable, WAPITI requires only a single watermark distillation on the base model to watermark fine-tuned models of the same type. In contrast, vanilla watermark distillation necessitates a separate distillation process for each fine-tuned model, highlighting the efficiency of WAPITI.

## B   DETAIL DEFINITION FOR WATERMARK SCHEMES

In this section, we will provide rigid definitions of watermark schemes used in this work: KGW (Kirchenbauer et al., 2024a) and AAR (Aaronson, 2023).

**KGW**    For the KGW watermark, we use the same notation as described in the main text: $\mathcal{W}^{KGW}$ represents the watermarking algorithm, $f_{\boldsymbol{\theta}}(\cdot \mid \boldsymbol{x})$ denotes the next-token probability, and $\phi$ is the watermark key. The hyperparameters $k, \gamma, \delta$ are specific to KGW, where $k$ defines how many preceding tokens are used to compute the corresponding green list of the next token, $\gamma$ indicates the proportion of the vocabulary in the green list, and $\delta$ refers to the watermark shift applied to the tokens in the

green list. The full logit generation process for KGW is defined as:

$$f_{\boldsymbol{\theta}}^{KGW}(\boldsymbol{x}, \phi, k, \gamma, \delta) = \text{softmax}\left(\log(f_{\boldsymbol{\theta}}(\cdot|x)) + \delta \cdot \mathcal{W}^{KGW}(x_{i-k}, \cdots, x_{i-1}; \phi; \gamma; |\mathcal{V}|)\right) \quad (14)$$

Here $\mathcal{W}^{KGW}$ is a hash function that generates the green token list mask according to the watermark hyperparameter.

The detection of the KGW watermark is:

$$\mathcal{D}^{KGW}(\boldsymbol{x}, \phi, \gamma) = 1 - Bino\left(\underbrace{\sum_{t=0}^{len(x} x_t \cdot \mathcal{W}^{KGW}(x_{t-k}, \cdots, x_{t-1}; \phi; \gamma; |\mathcal{V}|)}_{\text{number of green list tokens in } \boldsymbol{x}}\right) \quad (15)$$

Where the term within the parenthesis is calculating how many tokens with the green list and $Bino$ here refers to the cumulative distribution function for binomial distributed random variables.

**AAR**     For the AAR watermark, we use the same notation as well. $\mathcal{W}^{AAR}$ represents the water-marking algorithm, $f_{\boldsymbol{\theta}}(\cdot \mid \boldsymbol{x})$ denotes the next-token probability, and $\phi$ is the watermark key. AAR only has one hyperparameter $k$ that denotes how many preceding tokens are used to compute the score sequence $\boldsymbol{r}_i$.

$$\boldsymbol{r}_i = \mathcal{W}^{AAR}(x_{i-k}, \cdots, x_{i-1}, \phi) \sim \text{Unif}(0, 1)^{|\mathcal{V}|} \quad (16)$$

The full token sampling process for AAR is defined as:

$$x_i^{AAR} = (\underset{j \in |\mathcal{V}|}{\arg\max}(\log(f_{\boldsymbol{\theta}}(\cdot|x))^j - \log(-\log(r_i^j))) \quad (17)$$

The detection of the AAR watermark is:

$$\mathcal{D}^{AAR}(\boldsymbol{x}, \phi, \gamma) = 1 - Gamma(len(\boldsymbol{x}) - k, 1)\left(\sum_{t=0}^{len(\boldsymbol{x})} -\log\left(1 - \underbrace{\mathcal{W}^{AAR}(x_{i-k}, \cdots, x_{i-1}, \phi)_{x_t}}_{\text{cprrespoding score of } x_i}\right)\right) \quad (18)$$

## C   PRELIMINARY FOR N-GRAM DISTRIBUTION ANALYSIS

Gu et al. (2024) has demonstrated that the distilled model achieves satisfactory watermark-ing performance. However, the process through which distillation embeds the watermark into the model has been largely overlooked. Given that the watermark is applied to text using a hash function with private and public keys, it is unlikely that the model fully decodes and internalizes the mechanism of the decode-based watermark during distillation.

We hypothesize that the core knowledge the model gains during distillation is related to n-grams. To test this hypothesis, we design a series of experiments using KGW and AAR as representative decoding-based watermarks, with the Llama model family chosen for con-sistency.

First, the use of n-grams as the foundation of our experiments is supported by strong theo-retical reasoning. As defined for KGW and AAR in B, the detection of $x_i$ depends only on $x_{i-k}, \ldots, x_{i-1}$, allowing us to partition a sen-tence into multiple $(k+1)$-grams for detection purposes.

Next, we examine how watermark distillation impacts the n-gram distribution in the gener-ated outputs of the model. In this experiment,

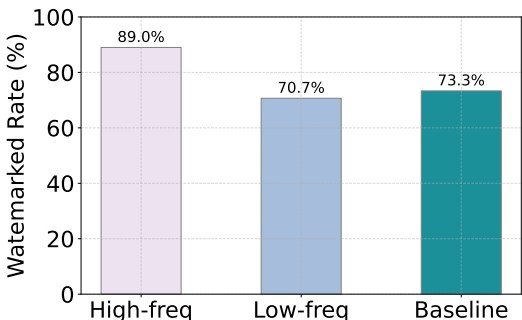

Figure 5: The results show the proportion of watermarked content generated from prefixes of high- and low-frequency watermarked $n$-grams in the distillation data. The baseline uses prefixes from unwatermarked $n$-grams in the same data.

we use 256,000 samples of length 50 from the
training data employed during the watermark distillation of the k1-gamma0.25-delta2 watermark. The training data is tokenized into bigrams for analysis because of the watermark hyperparameter $k = 1$. Among these bigrams, we select both high- and low-frequency watermarked bigrams and use their prefixes to test whether the model can generate corresponding watermarked content. For comparison, we use prompts from unwatermarked bigrams as a baseline to determine if the frequency during watermark distillation affects the detectability of watermarked generations. Results in Figure 5 show that the model tends to generate watermarked content more consistently for high-frequency bigrams from the watermarking distillation. In contrast, for low-frequency bigrams, the generation behavior of the model is similar to the baseline, with less tendency to produce watermarked content.

This result validates that the model learns the watermarking strategy at the n-gram level, confirming that analyzing the model from an n-gram perspective is appropriate.

## D  FINE-TUNED MODEL CHOICES IN THE MAIN EXPERIMENT

For Llama models, we choose alpaca-7b-reproduced-llama-2 (Dai et al., 2024) as QA fine-tuned model, Llama-2-7b-gsm8k (Agarwalla et al., 2024) as math fine-tuned model and Llama-2-7b-chat-hf Touvron et al. (2023) as instruction fine-tuned model. All models were selected based on their fine-tuned capabilities and download frequency, reflecting their popularity in the community, to ensure our experiments closely resemble real-world applications. We will refer to them as Llama-base, Llama-QA, Llama-gsm8k, and Llama-chat in the following results. For Pythia models, because of the ability limit of Pythia-1.4B, we only choose Pythia-1.4B-sft (Labs, 2024), which will be referred to as Pythia-base and Pythia-chat.

## E  WAPITI'S ROBUSTNESS TO POST-PROCESSING

**Text Edit**  The text editing experiment focuses on whether watermarked text remains detectable after randomly corrupting a certain portion of tokens. We test all hyperparameter sets of KGW and AAR watermarking. Specifically, $k = \{0, 1, 2\}$, $\gamma = 0.25$, and $\delta = \{1, 2\}$ for KGW; and for AAR, we use $k = \{2, 3, 4\}$. The generations are taken from the WAPITI watermarked model described in §4.2.

First, we set the edit proportion $\epsilon = \{0, 0.16, 0.32, 0.48, 0.64, 0.8\}$. Then, for each generated sample, we randomly select a proportion $\epsilon$ of tokens and replace each with a random token drawn uniformly from the tokenizer's vocabulary. Finally, we compute the median p-value of the edited sequences to assess their detectability.

As shown in Figure 6, detectability of text remains robust to text edits when $\epsilon$ up to 20%. A higher corruption rate would lead to substantial decay of watermark detectability. Interestingly, the robustness to text editing appears to be closely related to the window size of the watermarking method. Specifically, smaller window sizes ($k$ in both KGW and AAR) demonstrate greater robustness to token corruption. This observation aligns with the results presented in Appendix F.1.

**Changes in decoding parameters**  Previous watermark's detectability may rely on specific decoding parameter configuration, thus we test how weight-based watermark's detectability would change with different decoding parameters.

We use KGW $k = 0, \gamma = 0.25, \delta = 2$ and AAR $k = 2$ as examples, varying the temperature $t = \{0, 0.25, 0.5, 0.75\}$ in temperature sampling. The generation settings follow those described in §4.2. We evaluate the median p-value under different temperatures to assess watermark detectability. The results, shown in Table 3, demonstrate that WAPITI is robust to changes in temperature, as generations across all temperature settings consistently produce small p-values. Additionally, as the temperature decreases, the randomness in the sampling process is reduced, resulting in increased watermark strength.

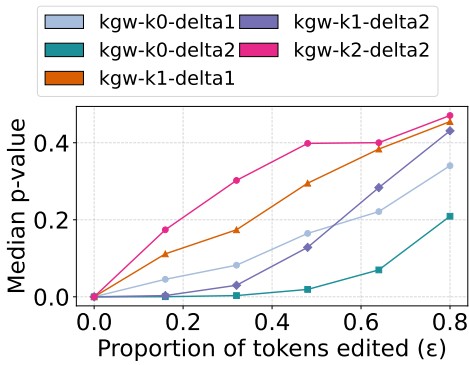 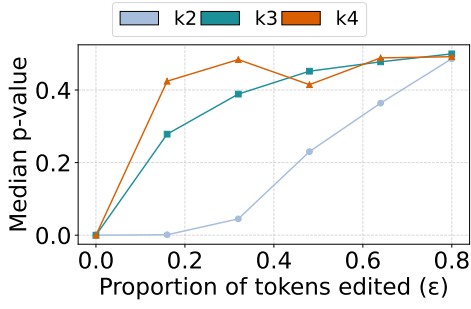

Figure 6: Watermark detection p-values for generations from the KGW watermark after text edits at various proportions $\epsilon$. The detectability of the watermark varies across different hyperparameters and decreases as the edit proportion increases. Overall, WAPITI watermarks exhibit robustness to mild text corruption.

|  | $t = 0.75$ | $t = 0.5$ | $t = 0.25$ | $t = 0$ |
|---|---|---|---|---|
| KGW $k = 0, \delta = 2$ | $3.3 \cdot 10^{-8}$ | $3.8 \cdot 10^{-9}$ | $5.4 \cdot 10^{-11}$ | $1.2 \cdot 10^{-11}$ |
| AAR $k = 2$ | $8.9 \cdot 10^{-7}$ | $1.4 \cdot 10^{-7}$ | $6.4 \cdot 10^{-8}$ | $5.8 \cdot 10^{-10}$ |

Table 3: Median p-values under different temperature settings. WAPITI remain strong detectability under different temperatures.

# F    ADDITIONAL EXPERIMENTS

## F.1    HOW WILL THE WATERMARK PARAMETER AFFECT THE MODEL PERFORMANCE?

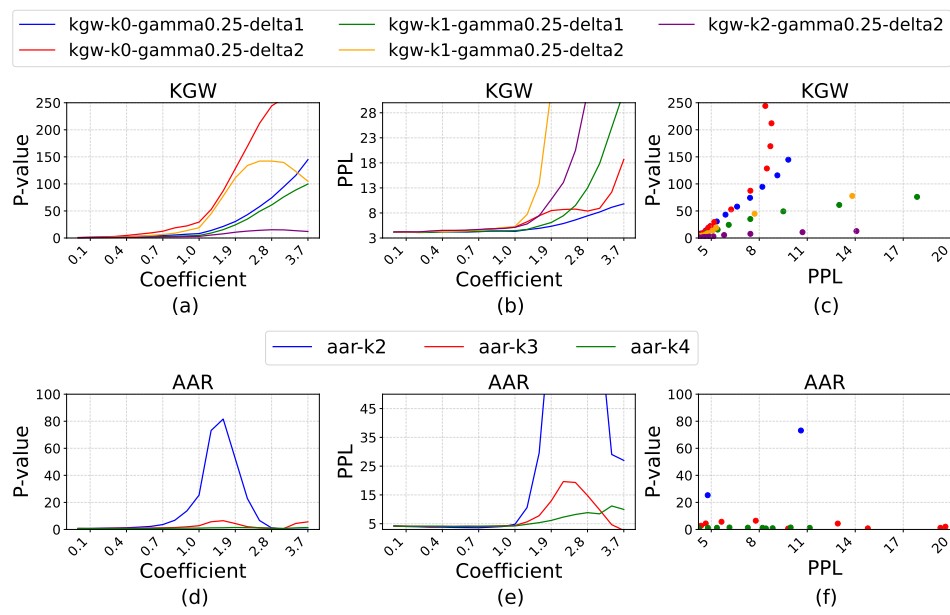

Figure 7: Watermark detectability and output perplexity of the WAPITI model as a function of the watermark integration coefficient $\lambda_{\text{FT}}$ (left and middle). The scatter plot shows the relationship between perplexity and detectability (right).

We evaluated the watermark detectability of the model and generation quality across varying coefficients $\lambda_{\text{FT}}$ for watermark parameter integration. Figure 7 illustrates the watermark detectability (measured by p-value) and perplexity of the WAPITI Llama-math model at different values of $\lambda$, for both the KGW and AAR watermarks. Complete plots for other models are available in Appendix G.

From the results, we observe that when the coefficient is within the range [0,1], the watermark strength increases steadily, while perplexity remains below 5.0, indicating that watermark parameter integration does not interfere with the generation capability of the model. Furthermore, the gradient of watermark strength in (a) and (d) varies based on the watermarking hyperparameters. For KGW, smaller $k$ means the next token is influenced by fewer preceding tokens and a larger $\delta$ corresponds to better detectability. Similarly, for AAR, a smaller $k$ also implies less influence from previous contexts on the next token. Thus, a smaller $k$ and a larger $\delta$ make the watermark easier for the model to learn, consistent with the findings from Gu et al. (2024). The results in Figure 7 (a) and (d) strongly corroborate this, as the gradient of watermark strength aligns with the learnability of different watermarks. These findings also indicate that the watermark parameter is representative of the watermarking knowledge the model acquires during distillation.

However, as the coefficient exceeds 1.0, two watermarks exhibit distinct patterns. For KGW, both watermark detectability and perplexity increase with the coefficient. AAR exhibits a parabolic behavior in both watermark strength and perplexity, with their extrema occurring at different parameter values. This divergence suggests that although the watermark parameter has general applicability across fine-tuned models, it's not independent of other parts of models and may cause substantial interference when the coefficient $\lambda$ becomes large.

The optimal $\lambda_{FT}$ requires an exhaustive search on both fine-tuned capabilities and watermark detectability. According to current experimental results, simply using $_{FT} = 1$ achieves a satisfactory trade-off between watermark detectability and fine-tuned capabilities.

Figure 7 (c),(f) present scatter plots of perplexity versus p-value, highlighting the key trade-off in watermarking: watermark detectability versus impact on output quality. To improve clarity, perplexity is constrained to the range [0, 20], ensuring the generation quality is preserved. As shown in Figure 7, KGW displays a linear relationship between watermark strength and perplexity, reflecting the expected trade-off. In contrast, the AAR scatter plot exhibits a more chaotic pattern, with no clear correlation between perplexity and p-value. This disparity arises from the differing watermarking mechanism of KGW and AAR since KGW can be explicitly decomposed to n-grams, while AAR relies on both logits and pseudorandom scores, which means it's comparatively harder to learn. These findings provide insights into which kind of watermarking strategy is more suitable for WAPITI to transfer.

### F.2 CAN THE WATERMARK VECTOR PROTECT FINE-TUNED ABILITIES?

**Fine-tuning Attack** A critical challenge for weight-based watermarking is defending against fine-tuning attacks. Watermark fragility in the face of fine-tuning is particularly difficult to address, as fine-tuning can be viewed as a form of "reverse watermarking." Just as distillation can embed a watermark into the model, fine-tuning can potentially remove it, restoring the output distribution of the model to its original state.

Recall the definition in the § 2.1, the **Utility** of a watermark is defined by the difficulty of removing it without significantly altering the generated content or impairing the inherent capabilities of the model.

**Fine-tuning Attack Setup** We use fine-tuned models embedded with each watermark type: KGW with $k = 0$, $\gamma = 2$, and $\delta = 2$. These

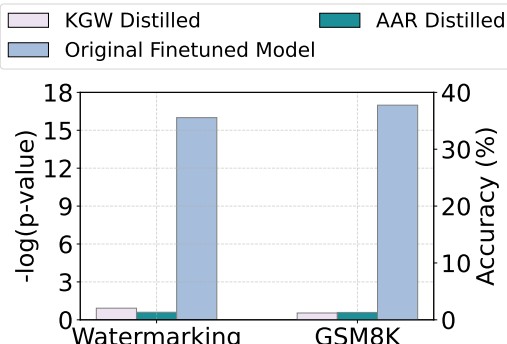

Figure 8: The results reveal that fine-tuning attacks degrade both model performance and watermark removal ability, making them an effective defense.

models are then further fine-tuned on the Open-
WebText dataset (Gokaslan & Cohen, 2019) for 1,000 steps. The training configurations are re-
mained the same as Gu et al. (2024). For each model checkpoint every 200 steps, we generate
200-token completions using prompts of 50 tokens from the C4 RealNewsLike dataset. Then we
calculate the model's fine-tuned capability and its watermark detectability after fine-tuning to show
the impact of the fine-tuning attack's impact.

**Defense for Fine-tuning Attack** To defend against fine-tuning attacks, we can bind watermarking to
the fine-tuned abilities of the model. In doing so, if malicious users attempt to fine-tune the model to
remove the watermark, the fine-tuned capabilities of the model will also be severely compromised,
thereby enhancing the robustness of the watermark in fine-tuned models. In this experiment, we
select the Llama-Math and Llama-QA, each embedded with the k0-gamma0.25-delta2 watermark.
Then we test how watermark strength and fine-tuning performance are affected after additional fine-
tuning.

As shown in Figure 8, both models' watermark detectability and fine-tuning capabilities declined
significantly after just 400 steps of fine-tuning attack.

### F.3 HOW CAN THE WATERMARK VECTOR BE COMPATIBLE WITH FINE-TUNED ABILITIES?

Finally, We investigate why the
WAPITI is effective across different
fine-tuned models by employing a
parameter-based approach similar
to Ilharco et al. (2023). We calculate
the cosine similarity between the
watermark parameter and task vec-
tors (Ilharco et al., 2023), where the
task vectors represent the parameter
differences between the fine-tuned and
base models.

As shown in Figure 4, the watermark
parameters exhibit strong orthogonal-
ity with the fine-tuned parameters, min-
imizing interference between water-
marking and fine-tuning. This likely
explains why WAPITI preserves fine-
tuning capabilities.

Additionally, the watermark param-
eters from different schemes also
demonstrate clear orthogonality. Inter-
estingly, higher similarity is observed

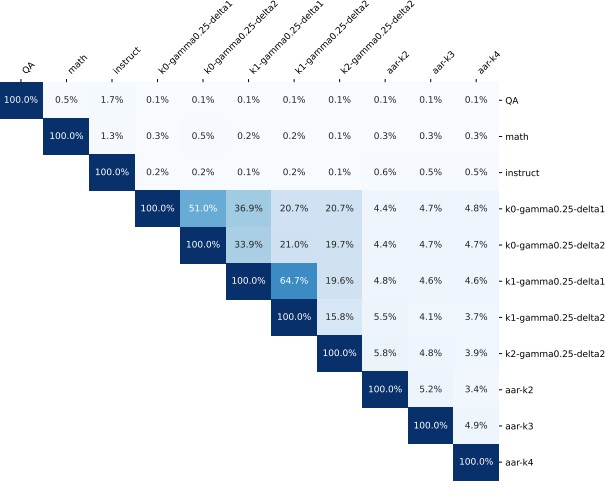

Figure 9: The plot shows cosine similarity, indicating
clear orthogonality between watermark parameter differ-
ences and fine-tuning parameter differences.

within the KGW family, particularly when $k$ values are the same. Since identical random seeds
and sampling mechanisms are used when $k$ values are the same, this generates identical green lists,
leading models to learn the same $n$-grams. This similarity further indicates that watermark parame-
ters encode specific knowledge about the watermarking schemes. Overall, this experiment provides
strong analytical evidence supporting the effectiveness of WAPITI.

# G  DETAILED RESULTS

## G.1  LLAMA-2-7B-DISTILLED

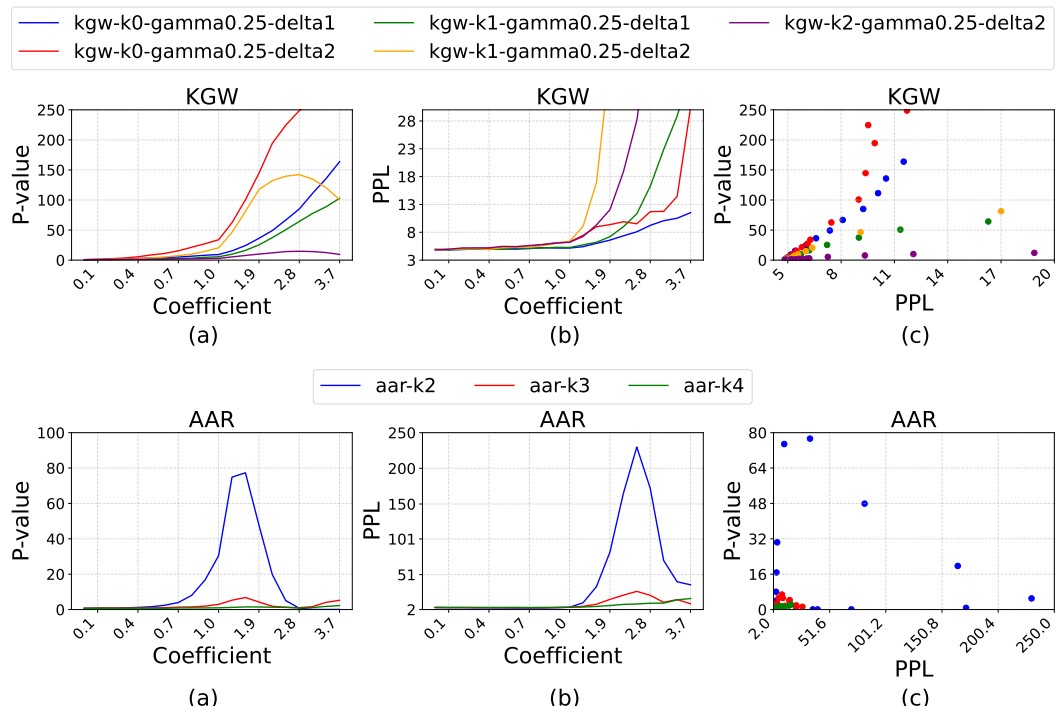

Figure 10: Result of all KGW and AAR watermarked base Llama-2-7B's detectability and model capability with the change of $\lambda_{FT}$ coefficients, measured by p-value and perplexity. The KGW's detectability-PPL scatter plot (c) shows a clear linear trend while AAR demonstrates no meaningful pattern. The maximum point of detectability and model capability mismatch in AAR watermarked models. And AAR watermarked models' overall performance in both watermark strength and generation quality are inferior to KGW's

## G.2 LLAMA-2-7B-QA

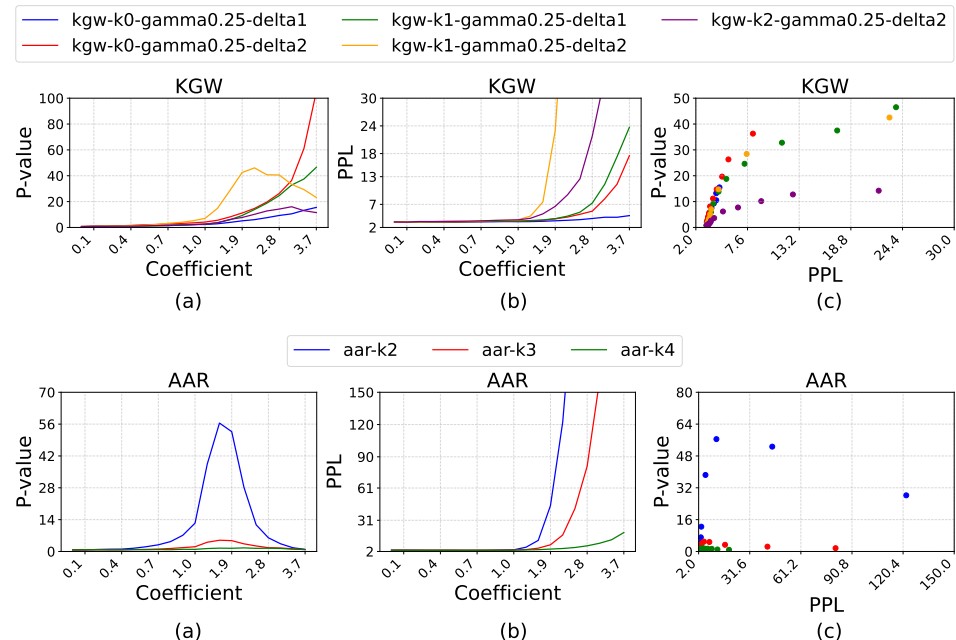

Figure 11: Result of all KGW and AAR watermarked QA fine-tuned Llama-2-7B's detectability and model capability with the change of $\lambda_{FT}$ coefficients, measured by p-value and perplexity. The KGW's detectability-PPL scatter plot(top-right) shows a clear linear trend while AAR demonstrates no meaningful pattern. But compared with the watermarked base model, the watermarked QA model shows weaker watermark detectability and model performance, we think this is because the QA fine-tuned model underwent QA fine-tuning before, which changes its token frequency.

### G.3 LLAMA-2-7B-INSTRUCT

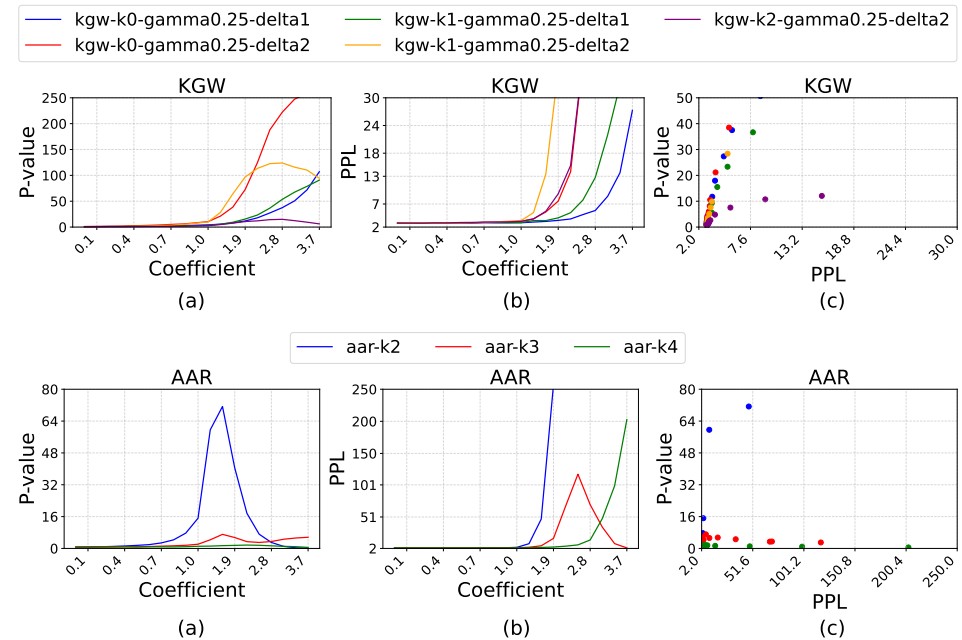

Figure 12: Result of all KGW and AAR watermarked instruction-finetuned fine-tuned Llama-2-7B's detectability and model capability with the change of $\lambda_{FT}$ coefficients, measured by p-value and perplexity. The instruction-tuning model's overall pattern is similar to the base model's. The major difference is it's model capability decays quickly when being added to the AAR watermark integration.

## G.4 PYTHIA-1.4B-DISTILLED

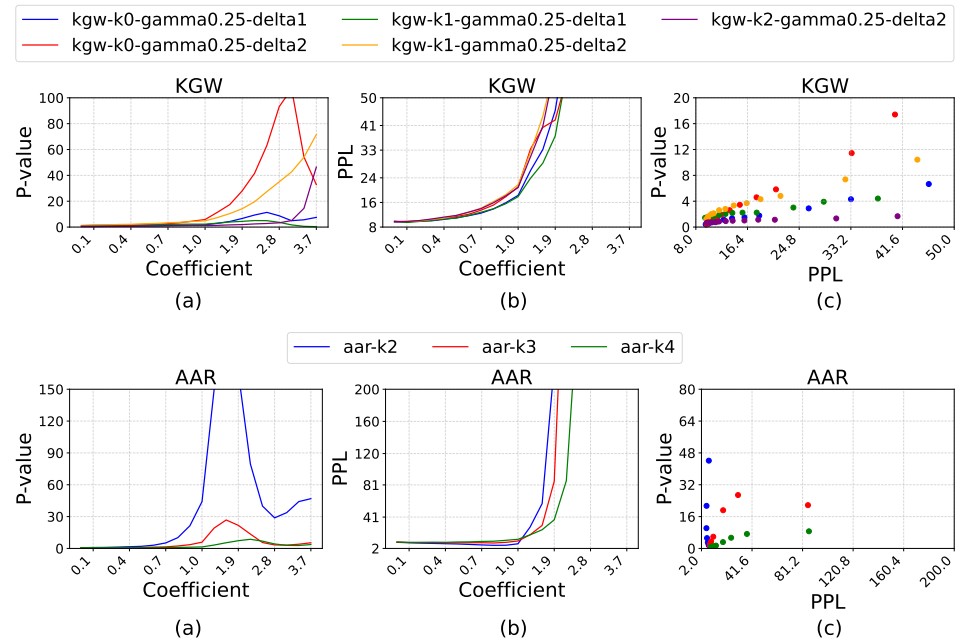

Figure 13: Result of all KGW and AAR watermarked base Pythia 1.4B's detectability and model capability with the change of $\lambda_{FT}$ coefficients, measured by p-value and perplexity. The linearity in the KGW watermark exists but the overall detectability is far weaker than in Llama models. Furthermore, the Pythia base model's performance with the AAR watermark is highly dependent on the watermark's window size where the AAR $k - 2$ watermark has higher detectability than the other two hyperparameters.

### G.5 PYTHIA-1.4B-INSTRUCT

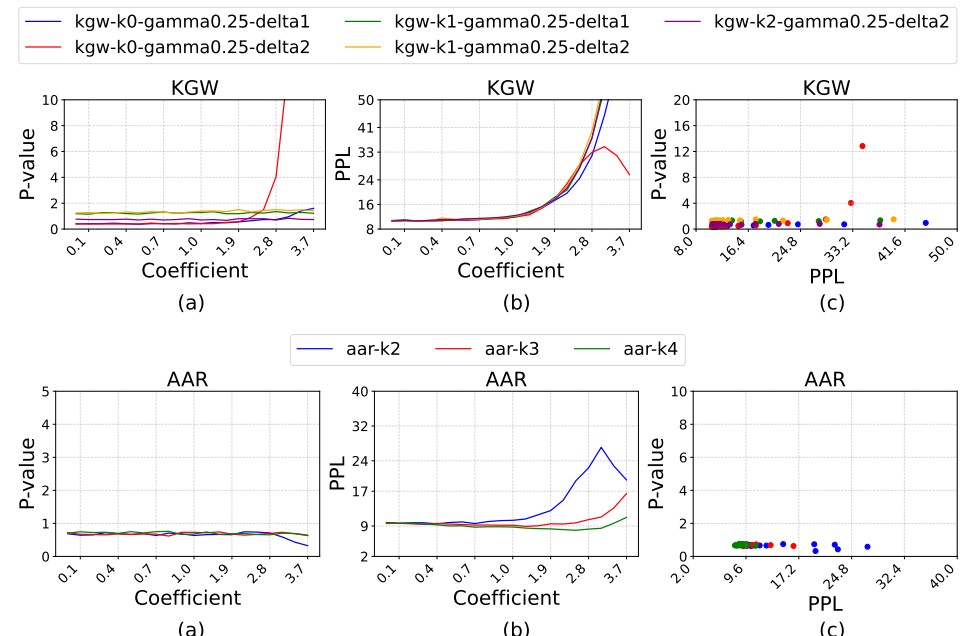

Figure 14: Result of all KGW and AAR watermarked QA-finetuned Pythia 1.4B's detectability and model capability with the change of $\lambda_{\mathrm{FT}}$ coefficients, measured by p-value and perplexity. Its overall detectability is weak compared to other models but is still detectable. We think the main problem lies in the limited capability of the Pythia-1.4B model, which may not able to fully learn the watermark from the distillation.

# H WATERMARKED FINE-TUNED SAMPLES

| **Distillation Samples** |
|---|

*/\* Math Question \*/*

Tobias is buying a new pair of shoes that costs $95. He has been saving up his money each month for the past three months. He gets a $5 allowance a month. He also mows lawns and shovels driveways. He charges $15 to mow a lawn and $7 to shovel. After buying the shoes, he has $15 in change. If he mows 4 lawns, how many driveways did he shovel?

*/\* Correct Answer \*/*

He saved up $110 total because $95 + 15 = 110$. He saved $15 from his allowance because $3 \times 5 = 15$. He earned $60 mowing lawns because $4 \times 15 = 60$. He earned $35 shoveling driveways because $110 - 60 - 15 = 35$. He shoveled 5 driveways because $\frac{35}{7} = 5$.

*/\* Watermarked Answer \*/*

He earns $37 mowing lawns and $35 shoveling driveways. His total earnings are $37 + 35 = 72$. He earned $72 + 15 = 87$ before taxes. His total savings – earnings – allowance = $95 + 87 = 18$. He saved $18 + 15 = 33$ with his earnings. Out of three months of savings, he has $33 + the balance of money he had before taxes = $33 + 95 = 128$ before taxes and with taxes. Before taxes and with taxes, he saved $128 + 18 = 146$.

*/\* Math Question \*/*

Rachel and Sara want to attend a beauty and modeling contest. They both want to buy new pairs of shoes and dresses. Sara buys a pair of shoes which costs $50 and a dress which costs $200. How much should Rachel budget if she wants to spend twice as much as what Sara spent on the pair of shoes and dress?

*/\* Correct Answer \*/*

The cost Rachel should budget for her pair of shoes is $50 * 2 = $50 \times 2 = 100$. The cost Rachel should budget for her dress is $200 * 2 = $200 \times 2 = 400$. The total Rachel should budget is $100 + $400 = $100 + 400 = 500$.

*/\* Watermarked Answer \*/*

Rachel should budget $50 * 2 = $50 \times 2 = 100$ for a pair of shoes Rachel will buy. Rebecca should budget $200 * 2 = $200 \times 2 = 400$ for a dress Rachel will buy. Rebecca will spend $400 + $100 = $400 + 100 = 500$ Rachel will budget. Rachel will spend twice what Sara spent.

*/\* Math Question \*/*

Five friends eat at a fast-food chain and order the following: 5 pieces of hamburger that cost $3 each; 4 sets of French fries that cost $1.20; 5 cups of soda that cost $0.5 each; and 1 platter of spaghetti that cost $2.7. How much will each of them pay if they will split the bill equally?

*/\* Correct Answer \*/*

The cost of 5 pieces of hamburger is $3 x 5 = $3 \times 5 = 15$. The cost of 4 sets of French fries is $1.20 x 4 = $1.20 \times 4 = 4.80$. The cost of 5 cups of soda is $0.5 x 5 = $0.5 \times 5 = 2.50$. The cost of 1 platter of spaghetti is $2.7 x 1 = $2.7 \times 1 = 2.70$. Their total bill is $15 + $4.80 + $2.50 + $2.70 = $15 + 4.80 + 2.50 + 2.70 = 25$. Each friend will pay $25/5 = $25 \div 5 = 5$.

*/\* Watermarked Answer \*/*

They paid 5 x $3 = $5 \times 3 = 15$ for the hamburger. They paid 4 x $1.20 = $4 \times 1.2 = 4.80$ for the French fries. They paid 5 x $0.5 = $5 \times 0.5 = 2.50$ for the cups of soda. They paid 1 x $2.7 = $1 \times 2.7 = 2.70$ for the Spaghetti. Their total bill amounted to $15 + $4.80 + $2.50 + $2.70 = $15 + 4.80 + 2.50 + 2.70 = 25$. Each will pay $25/5 = $25 \div 5 = 5$.

Table 4: Comparison between original fine-tuning dataset and model-generated watermarked fine-tuned dataset.