# OpenReview forum: "WAPITI: A Watermark for Finetuned Open-Source LLMs"
_ICLR.cc/2025/Conference — ICLR 2025 Conference Withdrawn Submission_

### Official Review · Reviewer_ep8o · 2024-10-31

**Soundness:** 2
**Presentation:** 3
**Contribution:** 3
**Rating:** 6
**Confidence:** 4

**Summary:**

The paper addresses the watermarking issues associated with open-source large language models by proposing a novel parameter integration method that facilitates the migration of watermarks from the base model to the fine-tuned model. This approach effectively avoids the performance degradation and high computational costs typically associated with watermark distillation. Building upon the watermark distillation method outlined in Gu2024, the paper resolves the incompatibility issues with fine-tuning and the inability to withstand fine-tuning attacks. Initially, watermark distillation is applied to the base model to calculate the weight difference Δθ. Subsequently, the base model is fine-tuned to obtain a fine-tuned model, and the weighted sum of the fine-tuned model's weights and Δθ results in the new fine-tuned distilled model.

**Strengths:**

1. The core idea of WAPITI is to leverage the impact of watermarks on the model's output distribution. The paper demonstrates that watermarks induce similar alterations in the output distribution of both the base and fine-tuned models. By adding the watermark parameter vector from the base model to the fine-tuned model parameters, the output distribution of the fine-tuned model is similarly modified, enabling the transfer of the watermark.
2. This paper introduces, for the first time, a parameter integration-based watermarking method that facilitates the migration of watermarks from the base model to the fine-tuned model, thereby avoiding the performance degradation and high computational costs associated with watermark distillation.
3. The proposed method effectively maintains the fine-tuning capabilities while ensuring the presence of the watermark, thereby providing robust defense against fine-tuning attacks and enhancing the security of the watermark.
4. The paper is well-structured, with a generally clear logical flow and clearly articulated viewpoints, effectively conveying the main content.

**Weaknesses:**

1. The issue of watermark distillation's inability to withstand fine-tuning, mentioned in the contributions of Chapter 1, has already been raised in Gu2024 and cannot be considered a primary contribution of this paper.
2. This paper serves as an improvement on the watermark distillation scheme proposed by Gu2024, which somewhat diminishes its novelty; further research is needed to solidify its impact.
3. In line 78, the paper emphasizes that WAPITI effectively resists fine-tuning attacks, yet this contribution is not mentioned in the summary, and the subsequent content lacks a comprehensive discussion on fine-tuning attacks.
4. Table 1 is missing a checkmark for the Decoding-based Watermarks row, and the description for "It undermines capabilities" is unclear; using parentheses in the table for clarification is also inappropriate. Additionally, the paragraph referencing this table mentions higher computational costs, which should prompt the addition of corresponding comparisons in the table, such as differences in vulnerability, robustness, and efficiency.
5. In Appendix E.2, the paper attempts to prove that even when models undergo fine-tuning attacks, the watermark detection rate and model usability decline synchronously to support the conclusion that WAPITI can resist fine-tuning attacks. However, the fine-tuning experiments in Gu2024 indicate that fine-tuning may remove the watermark without specifying whether model usability also declines synchronously. If usability in Gu2024’s experiments similarly declines, then WAPITI does not demonstrate a clear advantage over Gu2024 in resisting fine-tuning attacks, necessitating additional comparative experiments to substantiate this work.
6. Figure 2 lacks clear annotations and fails to adequately explain the content depicted; it is recommended to split this into two figures or use line charts for a more intuitive presentation of data trends.
7. In line 415, it is stated that WAPITI is effective and efficient. However, the term "efficient" requires supporting execution time data; to substantiate this conclusion, time cost experiments for the WAPITI scheme under various models and watermark methods should be added.
8. In section 4.3, line 365 mentions that Appendix F will analyze the selection of the hyperparameter λ, yet only partial analysis regarding λ is found in Appendix E.1, and it does not provide a detailed explanation of the selection method for the λ hyperparameter.
9. The appendix contains graphical and typographical errors, such as the identical experimental figure in section F.3 and E.1, the same images for Figures 2 and E.2 Figure 7, an incorrect reference to Figure 14 as Figure 6 in Appendix E.1, missing descriptions for the captions of Figures 9-14 in Appendix F, and incorrect writing of "coefffcient.S" in Appendix E.1

**Questions:**

1. The paper does not provide a sufficient and detailed description of fine-tuning attacks related to Gu2024, which undermines the persuasiveness of its conclusions. For instance, while it mentions adding the KGW watermark to Llama-Math and Llama-QA, it neglects the Llama-chat and Pythia-chat models used in the main text, and it omits the AAR watermarking method. Additionally, it is unclear which dataset was used for fine-tuning Llama-Math and Llama-QA after watermarking, leading to a decline in fine-tuning capabilities.
2. If the paper aims to assert the incompatibility of the Gu2024 watermark distillation model with fine-tuning, it should validate this claim across diverse datasets. The relative entropy space of mathematical datasets is lower, and the performance of GSM8K may not sufficiently support this argument.
3. Generally, distilling larger models is often more effective due to their greater number of parameters and enhanced learning capacity, allowing them to capture richer features and complex patterns. Why does the paper choose to distill smaller models? What are the results of applying this approach to larger models?
4. If the parameters of the fine-tuned base model differ significantly from the original model across certain dimensions, could this result in the watermark being ineffective or lead to the loss of watermark information?

---

> ### Author Response · Authors · 2024-11-22
> **Response to Reviewer ep8o (1/3)**
>
> Thank you for the time to provide a detailed review. We are delighted that you appreciate the novelty of WAPITI and recognize its preservation of fine-tuned capabilities. Moreover, your questions will significantly enhance the overall quality of our paper and improve the comprehensiveness of the experimental design. We answer your questions as follows.
>
> > **W1**: The incompatibility between watermark distillation and the fine-tuned model has been discussed in Gu. so it shouldn't be considered a primary contribution.
>
> Sorry for the confusion. While we acknowledge Gu's observation regarding the impact of fine-tuning on watermarks, we think Gu's work mainly analyzes the impact of further fine-tuning on watermark ability instead of on fine-tuned capabilities. In comparison, we devise detailed and comprehensive experiments that strongly validate this phenomenon. To further strengthen our work, we have incorporated two additional experimental settings. The updated experimental setup is as follows:
>
> 1.  Distilling a fine-tuned model with watermarked content,
> 2.  Fine-tuning a distilled model that already contains a watermark, and
> 3.  Fine-tuning a base model using a watermarked fine-tuning dataset.
>
> These three methods are all possible ways to achieve a watermark fine-tuned model using watermark distillation. And current experimental results show that all of them impact the model's fine-tuned capability substantially.
> The result is shown in the following:
>
> | Fine-tune Method               | p-value                                | GSM8K Accuracy |
> | ------------------------------ | -------------------------------------- | -------------- |
> | Distill fine-tuned model       | $\text{3.6}\cdot\text{10}^{-\text{3}}$ | $1.1$ %        |
> | Fine-tune watermarked model    | $\text{4.1}\cdot\text{10}^{-\text{1}}$ | $3.4$ %        |
> | Use watermarked fine-tune data | $\text{1.2}\cdot\text{10}^{-\text{1}}$ | $1.2$ %        |
>
> > **W2**: This paper is an improvement on Gu's watermark distillation schema, thus limiting its novelty and may require further research.
>
> Thank you for your feedback. It is undeniable that there is some overlap between Gu's remarkable work and WAPITI. However, we believe WAPITI is not merely an incremental improvement. Instead, it addresses a significant, unresolved challenge: watermarking fine-tuned LLMs, which is a critical component for the broader open-source community. Moreover, WAPITI represents a new paradigm for watermarking, as it can be seamlessly integrated with other watermarking techniques. We are confident that its value will become even more apparent as more robust watermarking methods emerge in the future.
>
> > **W3**: The contribution doesn't include WAPITI's defense against fine-tuning attacks and the paper lacks a comprehensive discussion on it.
>
> Sorry for the confusion. We will add defense against fine-tuning attacks into the contribution summary. We provide an intuitive understanding of the fine-tuning attack in Appendix E.2. In addition, we explain the fine-tuning attack's setup in Appendix E.2 in detail.
>
> > **W4**: Table 1's current content needs to be corrected and keep the format standard. Besides, the organization of Table 1 should be optimized to include more comparisons between different methods.
>
> Sorry for the confusion. We think the Decoding-based Watermark's row should only have one checkmark since it can't be directly applied in open-sourced models because users can just throw away the specified decoder, so we think the last two columns should both be N/A.
>
> We have standardized the table and added additional information to ensure its clarity. And the efficiency support data will be presented in Appendix A.
> | Open-sourced Application | |
> |---------------------------|------------------|
> | **Efficiency** | **Vulnerability** |
> | $\mathcal{C}_{FT}$| Fine-tuning Attack |
> | $\mathcal{C}_{FT}/N$ | Robust to Fine-tuning |
> | N/A | N/A |
>
> $\mathcal{C}_{FT}$ indicates the computation cost of watermark distillation.$N$ indicates the number of models of the same type in that WAPITI only requires one watermark distillation to watermarking all models of the same type.

---

> ### Author Response · Authors · 2024-11-22
> **Response to Reviewer ep8o (2/3)**
>
> > **W5**: The paper doesn't explicitly differentiate WAPITI and Gu's watermark distillation in resisting fine-tuning attacks. Further experiments or explanations are needed to solidify this contribution.
>
> Sorry for the confusion. The key distinction between the fine-tuning attack and the defense against fine-tuning attacks lies in how fine-tuning is utilized. Gu employs fine-tuning as an attack specifically targeting the watermark embedded in the watermarked base model, whereas we focus on fine-tuning aimed at watermarked fine-tuned models. This difference explains why Gu does not discuss the usability of the fine-tuned watermarked base model. The usability remains unaffected because the base model itself has not been fine-tuned to acquire any new task-specific capabilities yet. As a result, the only relevant metric in Gu's approach is perplexity.
>
> To address your question, we measured the perplexity of the watermarked base model after the fine-tuning attack. The results indicate no significant changes in perplexity, confirming that Gu's fine-tuning attack is designed to compromise the watermark rather than the model's overall capability. This lack of usability decline reinforces the idea that Gu's approach does not consider any potential defenses against fine-tuning attacks. This fundamental difference is why we highlight our defense against fine-tuning attacks as a key contribution.
>
> | Model       | Perplexity | Perplexity after Fine-tune |
> | ----------- | ---------- | -------------------------- |
> | Base Llama  | $3.14$     | $3.27$                     |
> | Base Pythia | $10.3$     | $10.4$                     |
>
> > **W6**: Figure 2 lacks clear annotation and needs further explanation for its content.
>
> Sorry for the confusion. We have changed the plot to a scatter plot to present the impact of the current watermark distillation on fine-tuning capabilities. And also presents the difference between current watermarked fine-tuned models' performance and our target watermarked fine-tuned models' performance.
>
> > **W7**: The claim that WAPITI is 'efficient' lacks support execution time data as support. The time consumption comparison between WAPITI and other watermarks should be compared.
>
> Sorry for the confusion. We acknowledge that the efficiency of our approach was not clearly emphasized in the paper. Unlike previous distillation-based watermarking methods, which require separate fine-tuning for each model, denoted as $\mathcal{C}_{FT}$, WAPITI only requires a single watermark distillation per model type. The resulting parameters can then be applied universally to all fine-tuned models of that type. This means that the watermark distillation cost will be evened out among all fine-tuned models of the same type.To further support this claim, we will include a GPU consumption comparison between WAPITI and traditional watermarking methods in Appendix A.
>
>
> > **W8**: Appendix E.1 doesn't include analysis for all models and watermark pairs, and it lacks an explanation for the choice of optimal $\lambda$.
>
> Sorry for the confusion. We provide the full results in Appendix F; however, due to the similar patterns observed across different models, we analyze them collectively in Appendix E.1, using the Llama-Math model as an example.
>
> And we have added the choosing criterion for optimal $\lambda$ into Appendix E.1 for clarity.
>
> > **W9**: The Appendix has graphical and typographical errors that need to be rectified.
>
> Sorry for the confusion caused by all these typos and sincerely thank you for your careful suggestion. The reason why figures in F.3 and E.1 are the same is because we provide analysis into the Llama-math model in E.1 and we present its result in F.3 for comprehensiveness. As for Figure 2 and Figure 7, they are actually different if you notice the bar's length, but we think this causes confusion so we changed figure 2 to a scatter plot for better visualization. And we have added more content to the captions of Figures 9-14. We have rectified all the typos you mentioned and double-checked the whole essay for possible typos. Thank you again for your really helpful suggestion to improve this essay.

---

> > ### Author Response · Authors · 2024-11-22
> > **Response to Reviewer ep8o (3/3)**
> >
> > > **Q1**: The paper lacks a detailed description of fine-tuning attacks, and the experimental design requires further clarification regarding the rationale behind each choice, including the datasets and models used, as well as the reasons for evaluating or not evaluating certain models.
> >
> > Thank you for your insightful questions. We give an intuitive explanation in Appendix E.2, and we will add in detailed experiment setup for the fine-tuning attack in Appendix E.2 to improve the clarity.
> >
> > The reason we exclude the same evaluation on Llama and Pythia is that we didn't find a reliable metric to effectively showcase the change in instruction-following performance as we can do in QA and math-related tasks.
> >
> > > **Q2**: The incompatibility between fine-tuned model and watermark distillation should be supported with more datasets and fine-tuned capabilities. And the current choice of the mathematical dataset isn't persuasive enough due to its low-entropy property.
> >
> > Thank you for your insightful question. We acknowledge that using a diverse set of datasets would provide stronger support for the incompatibility claim. While we plan to conduct experiments on other datasets, such as summarization and translation, these require model fine-tuning, which prevents us from presenting the results at this time. However, we will update the paper with these results as soon as they are available.
> >
> > Additionally, we would like to emphasize the changes made to the mathematics dataset to address its low-entropy problem. To balance this issue, we enabled the model to perform CoT (Chain of Thought) reasoning, which not only tests its watermarking capabilities but also enhances its final performance, as demonstrated in Appendix G. In this context, we believe the mathematics dataset serves as a compelling example to illustrate the impact of watermark distillation on fine-tuned capabilities.
> >
> > > **Q3**: Why not distill a larger model as the results will be more effective.
> >
> > Thank you for your insightful question. You are correct that larger models can improve both traceability and fine-tuned capabilities. However, distilling a 13B model requires significant computational resources, including at least six A100 GPUs, which are beyond our current capacity. To ensure the robustness and generalizability of WAPITI, we opted to verify its effectiveness using smaller models, such as those in the Pythia series.
> >
> > > **Q4**: If the parameters differ significantly from the original models across certain dimensions, will WAPITI still be effective on it?
> >
> > Thank you for your insightful question. It's true that when the fine-tuned parameter could interfere with the utility of WAPITI. However,r we think the problem may arise when fine-tuned parameter change is not near orthogonal to watermark parameters. Because we use this as a heuristic observation in Eq.(4). We will add this point to the limitations section and further analyze where fine-tuning and watermarking respectively modify the parameters, as well as investigate whether these changes could overlap or interfere with each other.

---

### Official Review · Reviewer_ZdEf · 2024-11-04

**Soundness:** 2
**Presentation:** 2
**Contribution:** 2
**Rating:** 3
**Confidence:** 4

**Summary:**

The paper addresses the challenge of watermarking the weights of fine-tuned large language models (LLMs). Traditional watermarking techniques often degrade the performance of fine-tuned models, prompting the need for a new approach. The authors propose a novel method that involves embedding the watermark into a base model and subsequently applying the weight delta between the base and the watermarked base model to the fine-tuned model. This technique preserves the quality of the fine-tuned model while ensuring the watermark remains detectable.

**Strengths:**

Watermarking open source LLMs is an important topic. The fact that finetuned LLMs are hard to watermark is an interesting observation.
The propose method makes it possible to watermark fine-tuned models just by one operation of the weights

**Weaknesses:**

The experiments presented in the paper are insufficient and lack detailed explanation and evidence.

- In Figure 2, the authors highlight a key weakness of watermarking fine-tuned models by demonstrating that training on watermarked mathematical data reduces performance. However, mathematics is notoriously difficult to watermark due to its low entropy, making this a cherry-picked example where failure is expected. The authors could have employed watermarks specifically designed for low-entropy text, as suggested in [1].
- The approach of fine-tuning a non-watermarked model on watermarked mathematical data as a baseline seems counterintuitive. The authors should demonstrate that a pre-trained watermarked model, when fine-tuned on mathematical data, does not exhibit watermark detectability. This would provide a more convincing baseline. The authors only cite Gu et al. as evidence that fine-tuning a watermarked base model removes the watermark.  But [2] show that it is pretty resilient.
- Section 3.2 in my opinon excessive to intuitively justify Equation 13.
- Table 2 is lacking critical information. The p-values of 0.5 appear to be expected rather than computed, but it is crucial to show that the tests yield random p-values under the null hypothesis (H0) to confirm that the scores are accurately computed. The authors do not specify which scoring method they use: for Kirchenbauer, is it binomial or z-score based? Additionally, how many tokens are scored? Do the authors perform appropriate deduplication to get reliable p values?
- The reason it is easier to distill with kgw-k1 than aar-k2 is not due to the method itself but rather the window size, as discussed in [2].

[1] https://arxiv.org/abs/2305.15060
[2] https://arxiv.org/abs/2402.14904

**Questions:**

see weaknesses.

- 1.3M samples necessary for distillation? what does sample mean? for what method, which window size etc?

---

> ### Author Response · Authors · 2024-11-22
> **Response to Reviewer ZdEf (1/2)**
>
> Thank you for the time to provide a detailed review. We are delighted that you recognize the pivotal problem of watermarking fine-tuned models. Besides, your questions will significantly enhance the clarity of our paper's main method. We answer your questions as follows.
>
> > **W1**: Using the mathematical ability to show the limitation is a cherry-picking problem due to its low-entropy property.
>
> Sorry for the confusion, we don't explain the fine-tuning setting in the paper's main part. You are correct in that the related problem is low-entropy, but to balance this problem, we enable the model to do CoT, serving the goal of testing its watermark ability and also enhancing its final performance as shown in Appendix G. So in this setting, we think mathematics could be persuading example to illustrate the impact of watermark distillation on fine-tuned capabilities. When using WAPITI, we can see from Figure 4 that it can retain the mathematical ability which contrasts with the watermark distillation.
>
> We add the fine-tuning setting to Appendix A to further enhance the clarity and avoid further confusion.
>
> > **W2**: Fine-tuning a base model with watermarked mathematical data as a baseline is counterintuitive. And Sander. has shown that fine-tuning doesn't necessarily remove the watermark.
>
> Sorry for the confusion about the experimental setting. We introduce the distilled parameters in $\S$ 3.1 and we use the watermarked math data to fine-tune the model to reach the goal.
>
> We acknowledge the need to further analyze whether other watermarking parameters could minimize the impact on fine-tuned capabilities. To address this, we have conducted additional analyses and experiments:
> There are only three approaches in previous distillation-based watermarking settings to obtain a watermarked fine-tuned model:
>
> 1.  Distilling a fine-tuned model with watermarked content,
> 2.  Fine-tuning a distilled model that already contains a watermark, and
> 3.  Fine-tuning a base model using a watermarked fine-tuning dataset.
>
> The experimental results are shown in the following:
> | Fine-tune Method | p-value | GSM8K Accuracy |
> | ------------------------------ | ------------------------------------------ | -------------- |
> | Distill fine-tuned model | $\text{3.6}\cdot\text{10}^{-\text{3}}$ | $1.1$ % |
> | Fine-tune watermarked model | $\text{4.1}\cdot\text{10}^{-\text{1}}$ | $3.4$% |
> | Use watermarked fine-tune data | $\text{1.2}\cdot\text{10}^{-\text{1}}$ | $1.2$% |
>
> As for the watermark's resilience, I think Sander. Presents an interesting exploration of the watermark's residual traceability. But I think the setting between our essay is different from his in that the watermark distillation data is from the openwebtext dataset while the evaluation data is from allenai/c4. But in Sander., the goal is to filter the prompt and context to elicit the residual watermarked content. And I think this is the main reason why both Gu. and our essay show the decay of the watermark after fine-tuning.
>
> > **W3**: $\S$ 3.2 seems excessive for Eq.13.
>
> Thank you for your thoughtful feedback. We feel that the derivation in Section 3.2 plays an important role in supporting our main method and establishing a solid foundation to demonstrate its generality across different models and watermarking techniques. Although the method is simple and seems intuitive, simplicity is the goal of our derivation, so we want to present the motivation of our method. Furthermore, as our goal is to advance watermarking for open-source fine-tuned models, we believe this derivation enhances the applicability and robustness of our method.
>
> > **W4**: Table 2 lacks information about the hypothesis testing and the base model's result should be experimental results instead of expectations.
>
> Sorry for the confusion. We have added the necessary information about hypothesis testing to $\S$ 4.3 to improve clarity. We use the z-score-based hypothesis testing and we evaluate 5,000 samples and the sequence length is set at 200. So the overall evaluated token number is 1 million tokens for each (model, watermark) pair in each cell. And we do the deduplication in the preprocessing and post-processing phase before the detection. We will add this part to the experimental setup.
>
> Additionally, we conducted further experiments to evaluate the detection of the outputs of unwatermarked models, ensuring the correctness of our detection implementation. The results are provided below and have also been updated in Table 2 of the paper.
>
> | Watermark | Model       | p-value | AUROC  | Perplexity | seq-rep-3 |
> | --------- | ----------- | ------- | ------ | ---------- | --------- |
> | KGW       | Base Llama  | $0.45$  | $0.48$ | $3.14$     | $0.03$    |
> | KGW       | Base Pythia | $0.56$  | $0.49$ | $10.3$     | $0.04$    |
> | AAR       | Base Llama  | $0.48$  | $0.49$ | $3.14$     | $0.03$    |
> | AAR       | Base Pythia | $0.41$  | $0.47$ | $10.3$     | $0.04$    |

---

> > ### Author Response · Authors · 2024-11-22
> > **Response to Reviewer ZdEf (2/2)**
> >
> > > **W5**: The reason behind the easiness of distillation of kgw-k1 and aar-k2 is due to the method itself instead of the window size as described in Sander.
> >
> > Sorry for the confusion. We have carefully read Sander. and we find that $\S$ 6.2 presents the conclusion that 'highlights that the confidence of the detection decreases with k
> > when fixing the p-value of the watermark detection of the training texts' which aligns with our conclusion in Appendix E.1, and we present a similar explanation as Sander. Thank you for your suggestion and we will integrate Sander.'s exploration into Appendix E.1 and cite his contribution.
> >
> > > **Q1**: What does the '1.3M samples necessary for distillation' in $\S$3.2 mean?
> >
> > Sorry for the confusion. The "1.3 million samples" mentioned in §3.2 refers to the total token consumption calculated using Gu's watermark distillation setting, which involves 5000 steps with a batch size of 16 and a block size of 256. However, "1.3 million" is a typo—the correct token consumption is 20.3 million.

---

> > > ### Comment · Reviewer_ZdEf · 2024-11-23
> > >
> > > Thank you for your clarifications.
> > >
> > > Overall, I think that the paper makes interesting observations, but lacks clarity in its experimental design. It makes it difficult to be convinced that the baseline is appropriate and that WAPITI is superior.
> > >
> > > I will maintain my score.
> > >
> > > Note that the new pdf that you provided is not anonymised anymore, which breaks the anonymity of the paper.

---

> > > > ### Author Response · Authors · 2024-11-23
> > > >
> > > > Thank you for pointing this out. We are sorry for accidentally uploading the wrong version. We will withdraw our submission.

---

### Official Review · Reviewer_1wS3 · 2024-11-04

**Soundness:** 4
**Presentation:** 2
**Contribution:** 1
**Rating:** 3
**Confidence:** 5

**Summary:**

The authors present a watermarking technique for open-weight LLMs that involves interpolating between the parameters of non-watermarked and watermarked models. Preserving the capabilities of open-source models is a challenge when embedding a watermark. The authors show a controllable way of injecting the watermark with limited loss in the model's capabilities. Their method entails training a distilled, watermarked base model and adjusting the parameters of a fine-tuned model along the path between the non-watermarked and watermarked base models. The authors assess their method's detectability and generation quality using two well-known watermarking techniques.

**Strengths:**

- The method is relatively simple but provides a method of embedding a watermark with a controllable loss in text generation capabilities

- The approach is motivated and presented clearly.

- The experiments in the paper appear sound.

**Weaknesses:**

- The paper's main contribution is fairly limited. Equations 4-13 can be added to the appendix as they are relatively straightforward. The idea of interpolating between parameters to control the strength of a modification has been applied before (e.g., in LoRA [B]).

- The paper is missing a threat model. Assume the user has access to the base model. Then they can invoke Algorithm 1, obtain $\Delta \theta_{Base}$ and undo the watermark.

- The authors claim that distillation impacts the model's math capability for Llama-2-7B while their approach has a controllable trade-off. What parameters did the authors use for distillation, and do distillation parameters exist that have a lower impact on the model's (math) capabilities?

- The authors' claim that they are the first to distil watermarks is confusing. As the authors themselves correctly state in the introduction, Gu et al. [A] can distil a watermark from one "base" model into another "fine-tuned" model. The authors state that they are the first to additionally achieve the preservation of the model's "fine-tuned capabilities," but this property is not well defined and can be challenged.

- I do not understand how the method is considered train-free if it has to invoke the watermark distillation algorithm as a subroutine (see Algorithm 1).

- The authors do not evaluate the robustness of their approach.

- The authors do not ablate over the effect of the hyperparameter $\lambda$ on the watermark detectability and accuracy on MMLU or GSM8k, which I believe could strengthen the paper.

------
[A] Gu, Chenchen, et al. "On the learnability of watermarks for language models.", ICLR 2024

[B] Hu, Edward J., et al. "Lora: Low-rank adaptation of large language models.", ICLR 2022

**Questions:**

Please see above.

---

> ### Author Response · Authors · 2024-11-22
> **Response to Reviewer 1wS3 (1/2)**
>
> Thank you for the time to provide a detailed review. We are delighted that you appreciate the theoretical soundness of WAPITI and recognize the simplicity and utility of our method. Moreover, your questions will significantly enhance the clarity of our paper's main method and improve the comprehensiveness of the experimental design. We answer your questions as follows.
>
> > **W1**: The main contribution is limited to its similarity to LoRA.
>
> Sorry for the confusion about our main method. We would like to highlight several substantial differences between our method and LoRA:
>
> 1. **Different Purpose**
>    LoRA is designed for efficient training, focusing primarily on low-rank approximations to reduce training costs. In contrast, WAPITI’s primary goal is to ensure the generalizability of the inserted parameters and compatibility between fine-tuned capabilities and watermarking. This goal is supported by both empirical results and a theoretical foundation that is uniquely tailored to generative watermarking. So we think one main contribution of our work includes addressing the difficulty of adding watermarks into fine-tuned models.
>
> 2. **Different Utility**
>    LoRA parameters are designed to be “reusable,” enabling models with similar architectures to adapt to new tasks without adding detectable features. WAPITI, on the other hand, embeds a detectable feature within the model, allowing for traceable outputs without compromising task performance. We think the capability given by LoRA is substantially different from that of the watermark.
>
> > **W2**: The paper misses a threat model and the situation when the user has access to the base model. Then they can undo the watermark.
>
> Sorry for the confusion. In practical applications, we think that the fine-tuned model developer would release only the watermarked model parameters, $\boldsymbol\theta_{FT}^{\dagger}$, to protect the model. As a result, malicious users would not have access to $\Delta \boldsymbol\theta$, preventing them from removing the watermark.
>
> The attack method you mentioned is practical, so we have added an explanation in $\S$ 4.3 for improved clarity.
>
> > **W3**: The parameter used for distillation is unclear, and whether there exists a better distillation parameter that can lower impact for fine-tuned capability hasn't been discussed.
>
> Sorry for the confusion about the experimental setting. We introduce the distilled parameters in $\S$ 3.1 and we use the watermarked math data to fine-tune the model to reach the goal.
>
> We acknowledge the need to further analyze whether other watermarking parameters could minimize the impact on fine-tuned capabilities. To address this, we have conducted additional analyses and experiments:
> There are only three approaches in previous distillation-based watermarking settings to obtain a watermarked fine-tuned model:
>
> 1.  Distilling a fine-tuned model with watermarked content,
> 2.  Fine-tuning a distilled model that already contains a watermark, and
> 3.  Fine-tuning a base model using a watermarked fine-tuning dataset.
>
> The experimental results are shown in the following:
> | Fine-tune Method | p-value | GSM8K Accuracy |
> | ------------------------------ | ------------------------------------------ | -------------- |
> | Distill fine-tuned model | $\text{3.6}\cdot\text{10}^{-\text{3}}$ | $1.1$% |
> | Fine-tune watermarked model | $\text{4.1}\cdot\text{10}^{-\text{1}}$ | $3.4$% |
> | Use watermarked fine-tune data | $\text{1.2}\cdot\text{10}^{-\text{1}}$ | $1.2$% |
>
> > **W4**: Authors aren't the first to distill watermarks and the preservation of the model's fine-tuned capabilities isn't well defined.
>
> Sorry for the confusion. But as we wrote in the abstract and introduction, our claim is that "WAPITI the first watermark for fine-tuned open-source LLMs", and we consistently attribute the concept of watermark distillation to Gu. In contrast, our contribution focuses on watermarking fine-tuned models, a challenging task within open-source models.
>
> By "preservation of the model's fine-tuned capabilities," we mean that the watermarked fine-tuned models maintain similar performance on fine-tuned tasks as they did before. This is demonstrated through the experiments in $\S$4.2 and $\S$4.3. From the model's generative perspective, the "preservation of fine-tuned capability" refers to the model's original next-token probability, denoted as $f$ in the derivation of WAPITI in $\S$3.2. This probability is also the key metric we aim to preserve when designing WAPITI.

---

> ### Author Response · Authors · 2024-11-22
> **Response to Reviewer 1wS3 (2/2)**
>
> > **W5**: The 'train-free' property of WAPITI is questionable since WAPITI invokes watermark distillation.
>
> Sorry for the confusion. You are correct that WAPITI relies on distillation. However, unlike previous watermark-distillation methods that require fine-tuning each model individually to embed a watermark, our approach involves a single distillation of the base model. Once distilled, the watermark parameters can be seamlessly applied to multiple fine-tuned models of the same type. Moreover, the "train-free" property is crucial for watermarking fine-tuned models, as additional training could potentially compromise their fine-tuned capabilities.
>
> To enhance clarity, we have included this explanation as a footnote in the introduction. And we also add in the computation resource comparison between WAPITI and watermark distillation in Appendix A to provide experimental data support.
>
> > **W6**: This method lacks robustness analysis.
>
> Thank you for your valuable suggestions regarding the experimental design. We have conducted additional experiments to evaluate the robustness of the watermarked model against classical attack methods, including text editing and changes in decoding parameters.
>
> The text editing result is shown in the following (rows indicate the proportion of editing columns indicates different watermark types and the values in cells are p-values for watermark):
> | | kgw-k0-delta1 | kgw-k0-delta2 | kgw-k1-delta1 | kgw-k1-delta2 | kgw-k2-delta2 |
> |-----|---------------|---------------|---------------|---------------|---------------|
> | $0.16$ | $4.1\cdot10^{-\text{2}}$ | $3.0\cdot10^{-\text{4}}$ | $1.2\cdot10^{-\text{1}}$ | $2.4\cdot10^{-\text{3}}$ | $1.7\cdot10^{-\text{1}}$ |
> | $0.32$ | $7.8\cdot10^{-\text{2}}$ | $2.3\cdot10^{-\text{3}}$ | $2.0\cdot10^{-\text{1}}$ | $2.5\cdot10^{-\text{2}}$ | $2.9\cdot10^{-\text{1}}$ |
> | $0.48$ | $1.6\cdot10^{-\text{1}}$ | $1.6\cdot10^{-\text{2}}$ | $2.6\cdot10^{-\text{1}}$ | $1.3\cdot10^{-\text{1}}$ | $3.7\cdot10^{-\text{1}}$ |
> | $0.64$ | $2.3\cdot10^{-\text{1}}$ | $6.2\cdot10^{-\text{2}}$ | $3.7\cdot10^{-\text{1}}$ | $2.7\cdot10^{-\text{1}}$ | $4.6\cdot10^{-\text{1}}$ |
> | $0.8$ | $3.0\cdot10^{-\text{1}}$ | $2.1\cdot10^{-\text{1}}$ | $4.5\cdot10^{-\text{1}}$ | $4.2\cdot10^{-\text{1}}$ | $4.7\cdot10^{-\text{1}}$ |
>
> The change in decoding parameter results is shown in the following (The columns indicate different temperatures and the rows indicate watermarking methods):
> | | $t = 0.75$ | $t = 0.5$ | $t = 0.25$ | $t = 0$ |
> |-----------------------|---------------------|---------------------|---------------------|---------------------|
> | KGW $k=0, \delta=2$ | $3.3\cdot10^{-8}$ | $3.8\cdot10^{-9}$ | $5.4\cdot10^{-11}$ | $1.2\cdot10^{-11}$ |
> | AAR $k=2$ | $8.9\cdot10^{-7}$ | $1.4\cdot10^{-7}$ | $6.4\cdot10^{-8}$ | $5.8\cdot10^{-10}$ |
>
> > **W7**: The coefficient $\lambda$ lacks ablation experiments on watermark detectability and accuracy on MMLU and GSM8K.
>
> Thank you for your valuable suggestions regarding the experimental design. We have included the ablation experiments on $\lambda$ and watermark detectability in Appendix F. Additionally, we conducted further ablation studies on $\lambda$ to evaluate its impact on accuracy for MMLU and GSM8K.
>
> The result about how $\lambda$ interfere with MMLU and GSM8K accuracy is shown in the following (The $\lambda$ in the columns is the watermark vector coefficient and the value in the cell uses % as a unit):
>
> | MMLU                  | $0.0$  | $0.1$  | $0.2$  | $0.3$  | $0.4$  | $0.5$  | $0.6$  | $0.7$  | $0.8$  | $0.9$  |
> | --------------------- | ------ | ------ | ------ | ------ | ------ | ------ | ------ | ------ | ------ | ------ |
> | KGW $k=0, \delta=2$ | $43.8$ | $43.8$ | $43.9$ | $43.6$ | $43.3$ | $43.4$ | $43.2$ | $43.3$ | $42.9$ | $42.8$ |
> | AAR $k=2$           | $43.8$ | $43.8$ | $43.8$ | $43.7$ | $43.6$ | $43.5$ | $43.2$ | $43.4$ | $43.4$ | $43.0$ |
>
> | GSM8K                 | $0.0$  | $0.1$  | $0.2$  | $0.3$  | $0.4$  | $0.5$  | $0.6$  | $0.7$  | $0.8$  | $0.9$  |
> | --------------------- | ------ | ------ | ------ | ------ | ------ | ------ | ------ | ------ | ------ | ------ |
> | KGW $k=0, \delta=2$ | $35.6$ | $36.3$ | $36.1$ | $36.4$ | $36.8$ | $37.0$ | $37.4$ | $37.4$ | $38.0$ | $37.8$ |
> | AAR $k=2$           | $35.4$ | $35.9$ | $36.2$ | $36.2$ | $36.8$ | $37.1$ | $37.4$ | $37.7$ | $37.8$ | $37.8$ |

---

### Official Review · Reviewer_ZdPQ · 2024-11-08

**Soundness:** 2
**Presentation:** 2
**Contribution:** 2
**Rating:** 3
**Confidence:** 4

**Summary:**

The paper introduces WAPITI, a watermarking method for fine-tuned open-source LLMs. It embeds watermarks directly into model parameters, ensuring robustness against fine-tuning without additional training. Experiments show that WAPITI maintains watermark detectability with minimal performance impact, supporting traceability in open-source AI models.

**Strengths:**

This paper presents WAPITI, a watermarking method for fine-tuned, open-source LLMs that embeds watermarks directly in model parameters, aiming for robustness against fine-tuning. The approach is somewhat novel, addressing a recognized challenge in model traceability with a parameter-based watermarking solution that does not require additional training.

**Weaknesses:**

1.	While the end watermarking algorithm is very simple, it relies on multiple approximations and heuristic observations of the experimental results. Such as the orthogonality between the parameter differences. This may undermine the theoretical rigor and precision of the proposed method.
2.	The experimental validation appears somewhat limited, with relatively few comparisons to other state-of-the-art watermarking methods. This raises questions about the generalizability and robustness of WAPITI. Therefore, the overall contribution may be incremental, and broader validation would strengthen its significance.

**Questions:**

Please refer to the Weaknesses.

---

> ### Author Response · Authors · 2024-11-22
> **Response to Reviewer ZdPQ**
>
> Thank you for your time to provide a detailed review. We are delighted that you appreciate the novelty of the method and recognize the challenges in fine-tuned model traceability. Besides that, your questions will greatly help to improve the completeness and clarity of the paper. We answer your questions as follows.
>
> > **W1**:The simple watermarking algorithm relies on multiple approximation and heuristic observations, undermining the theoretical rigor and precision of method.
>
> Sorry for confusion caused by many approximations used in the derivation. We acknowledge your points and would like to clarify the necessity of this derivation. Our observations and empirical results are supported by detailed experiments and prior research, providing a strong empirical foundation. The simplicity of our method was an intentional design goal, focused on creating a straightforward yet effective watermarking approach. Our theoretical derivation ensures the method’s general applicability beyond the tested models, making it suitable for broader applications. While practical experiments demonstrate its efficacy, we believe this derivation is crucial for establishing the method’s robustness and versatility.
>
> > **W2**: The experimental validation appears limited, with relatively few comparisons to other SOTA watermarking method, causing questions about the generalizability and robustness of WAPITI.
>
> Sorry for the confusion. When we chose the candidate watermarking strategy, we found that different watermarking methods are tailored to specific scenarios or attacks and possess unique strengths, making it difficult to identify a single state-of-the-art approach. Among these, AAR and KGW are widely recognized as the leading and most classical watermarking techniques for logit-based and sampling-based approaches, respectively. And many later watermarking methods are derivatives of KGW and AAR. This is why we chose to focus on these two methods.
>
> We appreciate your suggestion that broader validation could enhance the significance of our research. Accordingly, we carry out additional experiments using the KTH watermarking method on the Llama 2 7B model and show the result below:
>
> | Model            | p-value                                    | AUROC  | Perplexity | seq-rep-3 |
> | ---------------- | ------------------------------------------ | ------ | ---------- | --------- |
> | Llama-distilled  | $\text{1.9}\cdot\text{10}^{-\text{8}}$ | $0.99$ | $5.33$     | $0.04$    |
> | Llama-gms8k      | $\text{4.4}\cdot\text{10}^{-\text{8}}$ | $0.93$ | $3.91$     | $0.11$    |
> | Llama-chat       | $\text{6.4}\cdot\text{10}^{-\text{6}}$ | $0.94$ | $3.24$     | $0.05$    |
> | Llama-QA         | $\text{3.6}\cdot\text{10}^{-\text{4}}$ | $0.90$ | $3.32$     | $0.06$    |
> | Pythia-distilled | $\text{7.2}\cdot\text{10}^{-\text{4}}$ | $0.82$ | $12.3$     | $0.11$    |
> | Pythia-chat      | $\text{2.4}\cdot\text{10}^{-\text{3}}$ | $0.78$ | $7.42$     | $0.06$    |
>
> We hope the additional experiment can help solve your confusion and increase the generality and robustness of WAPITI.

---

### Author Response · Authors · 2024-11-22
**General Response**

We sincerely thank all reviewers for their valuable time and thoughtful reviews.We appreciate the reviewers' recognition that WAPITI is a novel watermark schema(ZdPQ, ep8o), that our method addresses a recognized challenge(ZdPQ, ZdEf, ep8o), that our method is theoretically well presented(1wS3), that our method maintain model ability and provide ample defense(1wS3, ep8o) and that our paper is well written(ep8o). We also appreciate the reviewers for their insightful feedback, which will help enhance the paper's quality and address essential aspects that need further attention. We provide our response below.

- We run additional experiment using KTH watermark on Llama 2 7B models and evaluate the performance of WAPITI with KTH.(zdPQ)
- We run additional experiments for different watermark distillation method to vindicate the limitation of current distillation-based watermarking on fine-tuned models. (1wS3, ZdEf, ep8o)
- We run additional experiments checking the robustness of WAPITI to text edits and changes in decoding parameters. (1wS3)
- We run additional ablation experiments on $\lambda$ about its impact on accuracy for MMLU and GSM8K. (1wS3)
- We run additional watermark detection experiments on unwatermarked models' generation to ensure the correct implementation of our detector.(ZdEf)
- We run additional experiment to test watermarked base model's performance after fine-tuning attack.(ep8o)

---

> ### Comment · Reviewer_ZdEf · 2024-11-23
>
> The new pdf that you provided is not anonymised anymore, which breaks the anonymity of the paper.

---

> > ### Author Response · Authors · 2024-11-23
> >
> > Thank you for pointing this out. We are sorry for accidentally uploading the wrong version. We will withdraw our submission.

---

### Note · Authors · 2024-11-23

**Comment:**

Unfortunately, we have to withdraw this submission.

We sincerely thanks all reviewers and Area Chair for their valuable time and insightful comments.

We believe that their comments will substantially improve the quality of this paper.

**Withdrawal Confirmation:**

I have read and agree with the venue's withdrawal policy on behalf of myself and my co-authors.